# Post-learning replay of hippocampal-striatal activity is biased by reward-prediction signals

**Emma L. Roscow** [1] ✉, **Timothy Howe**[1], **Nathan F. Lepora** [2,3] &
**Matthew W. Jones** [1,3]

Neural activity encoding recent experiences is replayed during sleep and rest to promote consolidation of memories. However, precisely which features of experience influence replay prioritisation to optimise adaptive behaviour remains unclear. Here, we trained adult male rats on a novel maze-based reinforcement learning task designed to dissociate reward outcomes from reward-prediction errors. Four variations of a reinforcement learning model were fitted to the rats' behaviour over multiple days. Behaviour was best predicted by a model incorporating replay biased by reward-prediction error, compared to the same model with no replay, random replay or reward-biased replay. Neural population recordings from the hippocampus and ventral striatum of rats trained on the task evidenced preferential reactivation of reward-prediction and reward-prediction error signals during post-task rest. These insights disentangle the influences of salience on replay, suggesting that reinforcement learning is tuned by post-learning replay biased by reward-prediction error, not by reward per se. This work therefore provides a behavioural and theoretical toolkit with which to measure and interpret the neural mechanisms linking replay and reinforcement learning.

Good decisions typically rely on past experience to guide future behaviour. Actions which have previously produced beneficial outcomes in a similar context can be reinforced to adapt behaviour for maximising benefit. The ability for brain activity to drive synaptic plasticity, establishing functional networks encoding and implementing task-relevant information and actions, is central to this learning. These functional networks are refined during sleep and rest, when many neurons switch to a so-called offline state in which they replay activity encoding previous or anticipated experiences rather than current events or behaviours[1–4]. This offline replay, found across cortical, limbic and basal ganglia regions, has been suggested to play roles in decision-making[5], emotional processing[6], generalising across episodes[7] and reinforcement learning[8].

Studies in which replay has been manipulated provide strong evidence for its contributions to memory consolidation. For example, artificially enhancing replay by presenting odours or sounds during sleep, which had previously been paired with object locations or visual stimuli, leads to better subsequent recall of the paired stimuli[9–12]. Disrupting replay events, meanwhile, impairs subsequent spatial memory[13–16].

An examination of how replay aids these cognitive processes requires assessment of which activity is replayed with greatest strength or frequency. Activity which is associated with experiences of reward[17–20] or fear[21,22], or with recent, repeated and/or novel experiences[23,24], is replayed preferentially. This suggests a replay bias towards the most salient experiences to be processed, consolidated or incorporated into an internal model of the world. However, these salient experiences could also be interpreted as those with the highest prediction error, i.e. the most unexpected and therefore informative experiences for updating internal models and for reinforcement

[1]School of Physiology, Pharmacology & Neuroscience, University of Bristol, Bristol, UK. [2]School of Engineering Mathematics and Technology, University of Bristol, Bristol, UK. [3]These authors contributed equally: Nathan F. Lepora, Matthew W. Jones. ✉e-mail: emma.roscow.research@gmail.com

learning. Tasks which involve learning the locations of rewards often conflate reward with reward-prediction error (RPE), leaving open the possibility that apparent replay biases towards reward actually reflect biases towards RPE.

Here we combine behaviour, reinforcement learning and electrophysiology to explore the hypothesis that RPE, rather than solely reward or salience, bias replay. We used variations of a reinforcement learning model, Q-learning, to estimate the value of actions encoded in the striatum during a reinforcement learning task, and varied the amount and type of replay in the model to predict behaviour. Reinforcement learning relies on inputs from hippocampus to ventral striatum[25–28], where representations of reward values differ following learning acquired over weeks compared to when acquired over minutes[29] and, correspondingly, reward-responsive cells are replayed preferentially in the ventral striatum[18]. We therefore propose that replay triggers value updates in the striatum, to enhance striatum-dependent reinforcement learning and moreover that activity encoding events that resulted in high RPE is preferentially replayed. To corroborate this, we also recorded single-unit activity simultaneously from the hippocampus and ventral striatum during learning of the same task, revealing signatures of inter-area reward prediction signals and intra-area reward-prediction-error signals being preferentially reactivated during post-task rest.

Q-learning[30] has been used successfully to model reinforcement learning, particularly in humans[31,32] but also in rodents[33–35]. Q-learning models fit both behavioural outcomes and striatal activity, suggesting that they describe mechanisms of updating values in the striatum in response to RPEs which in turn guide behaviour[36–39]. Temporal-difference-based RPEs, i.e. the difference between expected reward and actual reward which drives the update of $Q$ values, closely resemble the dopaminergic input of ventral tegmental area (VTA) to the striatum[39–41], which modulates synaptic plasticity in the striatum[42] and may provide a mechanism for the biological equivalent of Q-learning. Dyna-Q[43], a variant of Q-learning which incorporates offline temporal-difference updates, has been used to model replay in ways which produce learning qualitatively similar to animal reinforcement learning[44]. RPE-biased replay has also been incorporated into machine learning algorithms and shown to enable much more efficient reinforcement learning, including for Atari games[45] and navigating a simulated environment[46] faster and with more success compared to replay without such a bias[47]. These algorithms demonstrate the utility of prioritising replay by RPE, and provide a theoretical foundation for investigating RPE-biased replay in the hippocampal-striatal circuit.

We trained 6 rats on a stochastic reinforcement learning task which elicited both positive and negative RPE, and fitted Q-learning parameters to each rat's behavioural data. We then included replay events between sessions, to simulate the effect of replay during sleep on reinforcement learning. Four replay policies were compared, prioritising state-action pairs to be updated according to different biases: random replay, replay proportional to expected reward, and two forms of RPE-biased replay. Random replay was included as a control, while reward-biased replay reflects the prevailing view of how replay is prioritised. Fitting the model parameters showed that the two RPE-biased replay policies increased the model's predictive accuracy, while random and reward-biased replay did not. A separate cohort of 3 rats was trained on the same task while recordings were made in dorsal CA1 and ventral striatum. Pairs of CA1 and striatal neurons were reactivated within and between these regions during sharp-wave ripples in the post-task consolidation period. The most strongly reactivated cell pairs showed preferential firing during the approach towards a reward location with a high anticipated probability of reward, indicating replay of reward-prediction signals, not pure reward signals. Within the striatum, the most strongly reactivated pairs of striatal cells showed preferential firing following a less-expected reward, indicating replay of reward-prediction-error signals. This suggests that replay between

sessions of a probabilistic reinforcement learning task in rats is biased by RPE and not solely by reward.

## Results

### Rats successfully learned a stochastic reinforcement learning task

Six rats were trained to forage for stochastic sucrose rewards on a three-armed maze, to assess their reinforcement learning on a task where reward outcome and RPE were dissociable. Each arm was assigned as either high probability, mid probability or low probability, which determined the protocol for reward delivery (Fig. 1a). This was designed so that, once rats gained enough experience of the task to correctly anticipate the reward probabilities, receipt of reward would elicit a low RPE, medium RPE and high RPE on each arm, respectively. For the first 15 daily training sessions, the high-probability arm delivered a reward on 75% of legitimate arm entries, the mid-probability arm on 50%, and the low-probability arm on 25%. A legitimate entry was one in which a different arm had been entered on the previous trial; entering the same arm twice in a row was illegitimate and did not result in a reward delivery. For sessions 16–20, the difference in reward probabilities for the high- and low-probability arms was amplified: reward was delivered on 87.5% and 12.5% legitimate entries, respectively. For sessions 21–22 the reward probabilities for the high- and low-probability arms were switched, such that the (formerly) high- and low-probability arms delivered reward on 12.5% and 87.5% of legitimate entries respectively. This set-up meant that receiving a reward in a low-probability arm would elicit a higher RPE than the same reward value in a high-probability arm, so reward outcome and RPE could be dissociated.

Over 22 sessions, animals learned to distinguish between the high-, mid- and low-probability arms in their frequency of visits to each arm, indicating successful learning of the reward probabilities. Rats performed $45.1 \pm 2.5$ trials per session, eventually showing a significant preference for the high-probability arm and against the low-probability arm, evident by session 6 and stable by session 11. The six animals varied in the degree of their discrimination between the arms (Fig. 1b), but on average they distinguished between all arms on 14 out of 22 sessions (Fig. 1c; $\chi^2$ tests, uncorrected), visiting the arms which delivered a higher probability of reward more often, particularly in later sessions. To minimise the possible confound of the maze orientation in the room, the arm probabilities were rotated between animals (for example, animals may have shown a confounding preference for the arm which was closest to the door of the recording room).

To quantify performance on the task, each trial was coded as optimal or suboptimal according to the animal's choice given the arm most recently visited. Because no reward was given for re-entering the same arm consecutively, the optimal action choice following a visit to the mid- or low-probability arm was to visit the high-probability arm; the optimal action following the high-probability arm was the mid-probability arm. Over sessions, animals increased the proportion of trials on which they behaved optimally, achieving performance significantly above chance level of 33% from session 3 onwards (one-sided binomial tests, Bonferroni-corrected). Using a more conservative chance level of 50%, to account for rats' natural tendency to alternate rather than repeat arms, they performed significantly above chance on 8 out of 22 sessions (Fig. 1d).

Reward probabilities were changed twice over the course of learning, triggering clear changes in behaviour. In the revaluation learning stage (sessions 16–20), the reward probabilities at each arm became more distinct: the high-probability arm delivering an 87.5% probability of reward compared to 75% in the initial learning stage, and the low-probability arm delivering a 12.5% probability of reward compared to 25% in the initial learning stage. This change offered a higher incentive-to-cost ratio and, correspondingly, preference for the high-

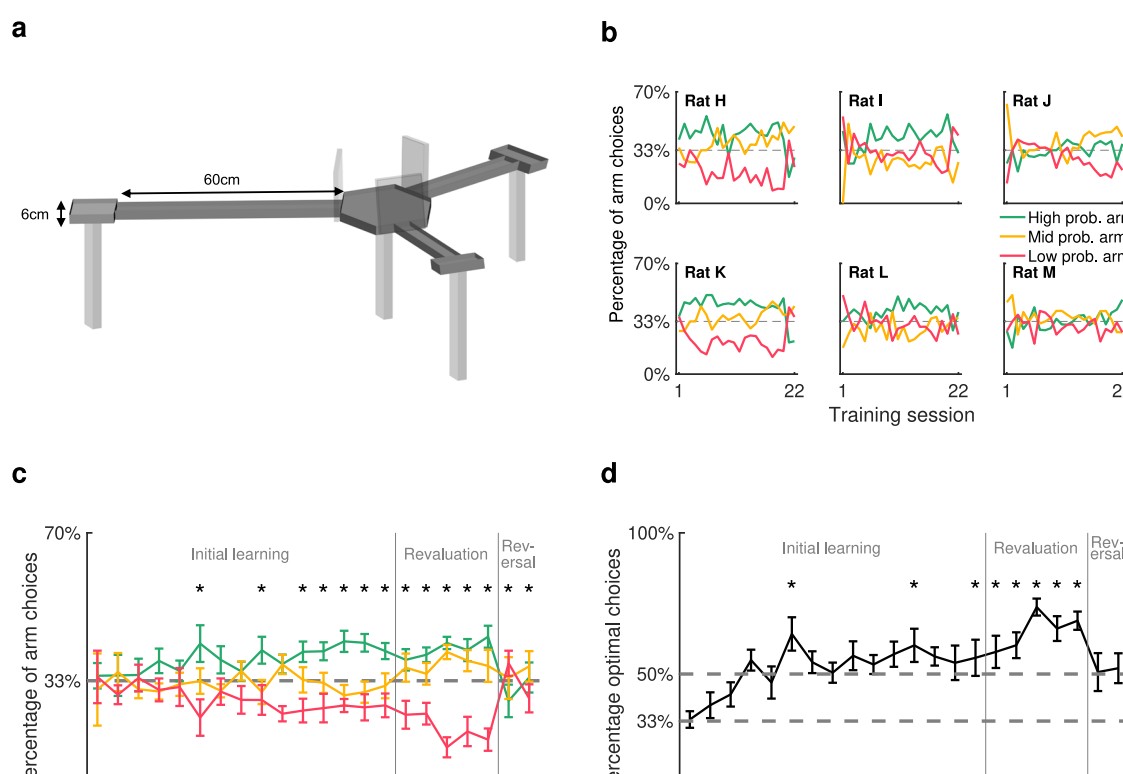

**Fig. 1 | Behavioural performance on the task. a** Illustration of the maze used to train animals. Lick ports located at the end of each arm delivered reward with either high, medium or low probabilities. **b** Frequency of entry to each arm over all sessions, shown separately for each rat. **c** Frequency of entry to each arm averaged across the 6 rats. Dashed line represents chance level (33.3%). * indicates arm choices statistically different from each other ($\chi^2$ test, $p < 0.05$, uncorrected). Error bars represent standard error of the mean (s.e.m.). **d** Mean proportion of trials pooled from 6 rats on which the optimal arm was chosen, according to highest probability of reward. Dashed lines represent chance levels (33.3% and 50.0%). * indicates performance statistically above 50% (one-sided one-proportion z-test, $p < 0.05$, Bonferroni-corrected). Error bars represent s.e.m. Source data for (**b**–**d**) are provided as a Source Data file.

probability arm over the low-probability arm increased compared to the previous five sessions (Fig. 1c; repeated-measures ANOVA, $F(1) = 9.37$, $p = 0.005$). As a result, the rate of optimal performance was also greater in the revaluation stage than during the last five sessions of the initial learning stage (Fig. 1d; repeated-measures ANOVA, $F(1) = 13.2$, $p = 0.001$).

The definition of optimal behaviour was the same in the initial and revaluation learning stages, because the arms did not change. However, optimal behaviour required a different behavioural policy in the reversal learning stage (sessions 21–22) when the high- and low-probability arms were switched. As expected, optimal performance correspondingly dipped when reward probabilities were reversed in sessions 21–22 as this new behavioural policy was learned: the frequency of optimal arm choices during the reversal learning stage fell to roughly the 50% chance level. These behavioural data demonstrate that reward probabilities successfully influenced learning and behaviour in the task, and that animals were capable of showing flexibility in response to changing reward. We therefore went on to test whether reinforcement learning algorithms were able to recapitulate rat behaviour and whether instantiating between-session (offline) replay of different task features improved model performance.

**Q-learning modelled animal behaviour**
We trained a Q-learning algorithm with no replay to generate probabilities of each action for each trial, based on $Q$ values estimated from the animals' previous experience (Fig. 2). Q-learning is a reinforcement learning algorithm in which an agent selects actions in its environment

and observes the outcome, recording at each time step $t$ its starting state $s_t$, selected action $a_t$, resulting reward $r_t$ and resulting state $s_{t+1}$. The agent builds up a matrix $Q$ of $Q$ value estimates for every state-action pair:

$$\begin{bmatrix} Q_{s_1, a_1} & Q_{s_1, a_2} & \cdots & Q_{s_1, a_A} \\ Q_{s_2, a_1} & Q_{s_2, a_2} & \cdots & Q_{s_2, a_A} \\ \vdots & \vdots & \ddots & \vdots \\ Q_{s_S, a_1} & Q_{s_S, a_2} & \cdots & Q_{s_S, a_A} \end{bmatrix} \quad (1)$$

corresponding to the future discounted expected reward, i.e. the temporal difference between the current state and the reward state. These $Q$ value estimates are used to guide actions to maximise reward. At each time step $t$, the $Q$ value for the state-action pair observed is updated by:

$$Q(s_t, a_t) \leftarrow (1 - \alpha) \cdot Q(s_t, a_t) + \alpha \cdot (r_t + \gamma \cdot \max Q(s_{t+1}, a)) \quad (2)$$

where $\alpha \in (0, 1)$ is a learning rate parameter which determines the degree to which new information overrides old information, and $\gamma \in (0, 1)$ is a discount parameter which determines the importance of long-term gains. In this task, entries into a chosen arm (and arrival at the goal location at the end of the arm) were modelled as actions, while the arm entered on the previous trial, on which reward probabilities were

**a**

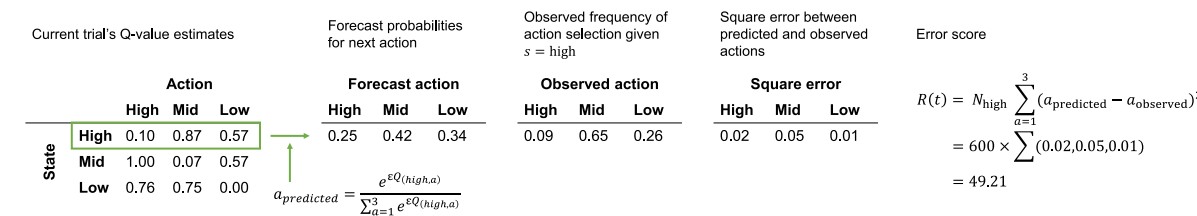

**b**

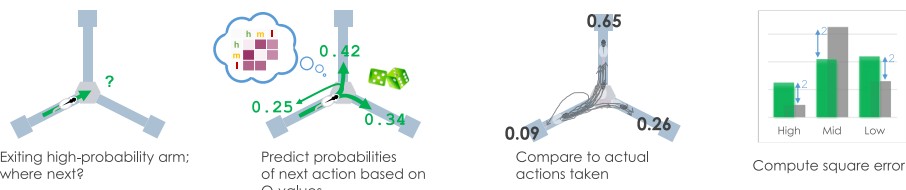

Exiting high-probability arm; where next?

Predict probabilities of next action based on Q-values

Compare to actual actions taken

Compute square error

**Fig. 2 | Example of model prediction for one trial, t = 100, in which rat H had most recently visited the high-probability arm (s = high) and chose the mid-probability arm (a = mid). a** The far left table shows the Q-learning model's estimate of the $Q$ values based on rat H's experience to date. Other tables show the predicted action probabilities calculated from the $Q$ values, the ground-truth of observed action frequencies over all visits to this state, and the mean square error between them. Far right shows how the error for this trial is calculated. **b** A cartoon illustration of the same trial: $Q$ values are used to predict action probabilities (green), the action frequencies are observed for the current state (grey) and the error score is computed from their squared difference.

contingent, were modelled as states. Each trial, therefore, gave rise to one state-action transition out of nine possible state-action pairs.

For each trial, a matrix of $Q$ values for all state-action pairs was updated based on experience and used to calculate predicted action probabilities, which were compared to the observed frequencies of state-action pairs to produce a vector of errors for the three available actions. An error score was calculated from the summed square of the error vector, weighted by the prevalence of the state. This produced a measure of how reliably the $Q$ value estimates predicted behaviour (Fig. 2; see 'Methods').

Observed action frequency correlated well with predicted action probabilities (Fig. 3a), indicating a good baseline model for reinforcement learning. Predicted action probabilities were binned in 100 percentile-bins for each animal, and for each bin the average frequency of these actions occurring was compared to the average predicted probability, resulting in a strong correlation ($R^2 = 0.87$, $p < 0.0001$, linear mixed-effects model). While individual rats alternated between arms on 94–96% of trials, the Q-learning agents fitted to each rat's behaviour alternated between arms on 92–95% of trials.

The error between predicted action probability and observed action frequency spanned a large range, which was greatest in the earlier training sessions and diminished towards 0 for later training sessions as $Q$ values were learned (Fig. 3b; early trials in blue have larger errors).

Error scores spanned a different range for each rat (Fig. 3c), so all further analysis was performed on error scores normalised by the mean for each animal. On this measure, normalised error was similarly highest in early training sessions, when behaviour is least optimal and most unpredictable. Following this, error became consistently low for most sessions (Fig. 3d), confirming a consistent fit with behaviour which captured the learning process over multiple sessions and changes in reward probabilities.

As described in Methods, the error score was used as the cost function to optimise three parameters in the Q-learning algorithm for each animal: a learning rate $\alpha$, a discount factor $\gamma$, and an exploration factor $\epsilon$. The resulting optimised parameter values are shown in Table 1. A perturbation analysis was performed to verify that the Q-learning results were sufficiently insensitive to perturbations to the optimised parameter values. At the optimised values, the average normalised error over all trials was, by definition, 1. Perturbing these values by up to 50% in either direction increased the normalised error by less than 0.5 in most cases (Fig. 3e), indicating that error score was not overly sensitive to small changes in parameter values. This confirms that the optimised models converged to a stable minimum that robustly captures rats' behaviour.

The model makes a simplifying assumption of stationary parameters throughout learning, which may deviate from biological reality[48] but prioritises interpretability of the fitted parameter values and prevents overfitting to an overly complex model.

In summary, the Q-learning algorithm proved able to recapitulate rat behaviour over the course of training and adaptation to new task conditions. The model was robust across a range of parameter values and established a sound basis on which to quantify the effects of simulating replay by updating $Q$ values between sessions.

**Adding RPE-biased replay to the Q-learning model improved prediction accuracy over reward-biased and random replay**

Against the baseline of no-replay, a variant of the Q-learning algorithm with replay was trained on the same data, with a specified number of samples chosen from all the trials experienced so far to be replayed between each session. Q-learning parameters were optimised for a fixed ($1 \leq n \leq 100$) number of replay events between each session, for each replay policy. All trials experienced by the animal were stored in a memory buffer, and for each replay event a state-action pair was chosen according to the replay policy and a sample trial from this state-action pair was used to update its $Q$ value (Fig. 4). The policies were defined as follows:

- With a random replay policy, all state-action pairs that had been experienced were sampled at random.
- With a reward-biased replay policy, state-action pairs were sampled in proportion to their $Q$ values, so that state-action pairs

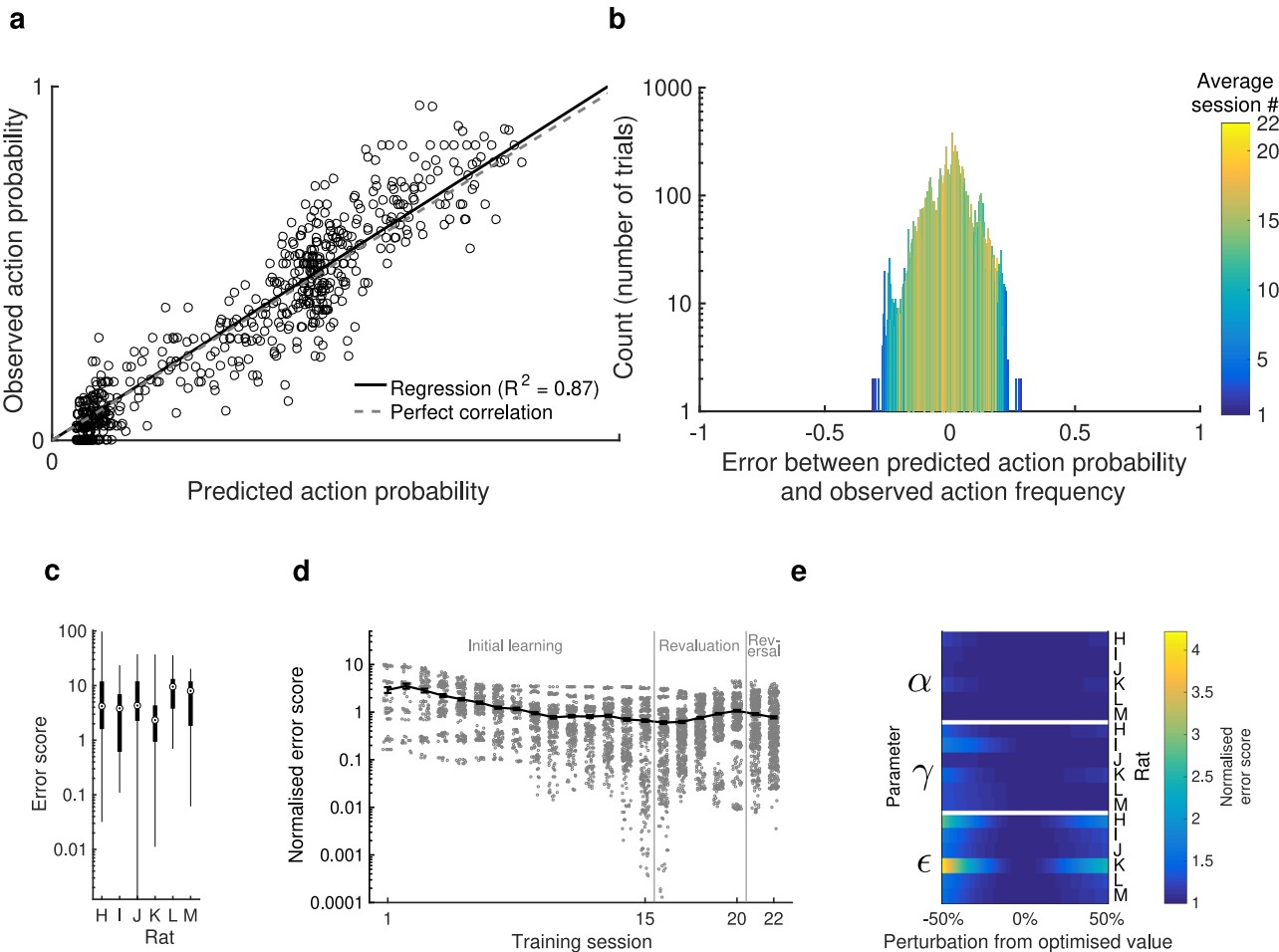

**Fig. 3 | Goodness of fit of the optimised Q-learning parameters, with no replay.** **a** Reliability diagram (trials pooled across all animals). Observed action probability indicates how often an action was chosen by the animal, averaged over similar predicted action probabilities. Data points represent per-rat percentile averages of action probabilities. **b** Histogram of residuals of the data in (**a**). Colour scale indicates, on average, which session the residuals within each bin occurred in. **c** Range of error scores for each trial (calculated from residuals) for each animal. An error of 0 reflects perfect modelling of action choices. Boxes represent 25th and 75th percentiles, circles represent median, whiskers represent range. $n = 603–1394$ error scores per rat. **d** Error scores pooled across rats and grouped into training sessions, normalised to the average error for each animal (shown in Table 1). $n=55$-523 error scores per session. Data points show normalised error for all trials; solid line represents mean for all animals. Error bars represent s.e.m. **e** Change in error score, normalised to the optimised error score for each animal, with varying perturbations to the optimised parameter values. The optimised values for learning rate $\alpha$, discount factor $\gamma$ and exploration factor $\epsilon$ were individually perturbed by 1–50% above and below the optimised value and the Q-learning algorithm was trained on behavioural data according to the perturbed parameter values 1000 times to obtain an average. Source data for (**a–e**) are provided as a Source Data file.

at which rewards had been experienced most frequently would be replayed most.
- With an RPE-prioritised replay policy, the state-action pair with the highest recent average RPE was sampled.
- With an RPE-proportional replay policy, state-action pairs were sampled in proportion to their recent average RPE.

The latter two policies offered two variations on preferentially updating state-action value(s) which had generated the greatest errors, concentrating efforts on correcting the most inaccurate expectations of reward (Fig. 4).

Compared to the no-replay Q-learning baseline, only replay which prioritised the highest-RPE state-action pair produced a more reliable model of learning (Fig. 5a; purple; linear mixed-effects model, two-sided), which was statistically significant even with one sample replayed between sessions. RPE-proportional replay produced a model which was numerically better but did not reach statistical significance (Fig. 5a; orange), while replay that was random or biased by reward did

not produce a more reliable model (Fig. 5a; blue and green). Replay of information encoded during trials associated with the most unexpected outcomes therefore significantly improved learning in the model, whereas replay of rewarded trials did not. This was true for all subjects: for 4 out of 6 rats the RPE-prioritised replay policy gave the lowest error, and for 2 out of 6 rats the RPE-proportional policy gave the lowest error (at 100 samples replayed for each policy).

The superiority of the RPE-prioritised replay policy was not uniform over the whole training period, however. With 100 replayed samples, all replay policies showed some modest improvement over no-replay in early sessions (Fig. 5c), but this effect disappeared in the random and reward-biased policies after roughly the seventh session. Conversely, the superiority of RPE-prioritised replay persisted over the whole course of learning. In the no-replay baseline, error scores increased in sessions 17–20. This reproduces an increase in optimal behaviour in these sessions during the revaluation stage and reversal stage respectively, suggesting that the model failed to capture subtleties in the learning pattern at these points when animals were adapting their behaviour to changes in reward probabilities. As

animals re-evaluated the state-action pairs in sessions 17–20 and adjusted their behaviour accordingly, replay by any policy was sufficient to overcome the increase in error scores seen in the baseline, so there was no increase at these sessions (Fig. 5c). This may reflect the faster learning enabled by replaying recently experienced trials. However, as animals reversed their behaviour in session 22, requiring a substantial update to $Q$ values and a dramatic change in behaviour, increased random replay or reward-biased replay did not improve error scores. Figure S1 shows an example of how $Q$ values were updated more rapidly with RPE-prioritised replay than random or reward-biased.

### RPE-biased replay did not improve predictions when trained on shuffled data

Given the indication that replay might play different roles in different learning stages, it is important to control for the possibility that parameter values were optimised for the general statistics of rewards

**Table 1 | Optimised parameter values for Q-learning algorithm trained on each animal's behavioural data**

|       | α      | γ      | ε      | Error score |
|-------|--------|--------|--------|-------------|
| Rat H | 0.0111 | 0.6805 | 2.6444 | 10.1367     |
| Rat I | 0.0132 | 1.0000 | 2.5555 | 5.1981      |
| Rat J | 0.0026 | 1.0000 | 2.7749 | 9.2751      |
| Rat K | 0.0319 | 0.6130 | 2.5299 | 3.7080      |
| Rat L | 0.0036 | 1.0000 | 2.2478 | 10.416      |
| Rat M | 0.0038 | 1.0000 | 2.6368 | 7.7669      |

α is the learning rate, γ is the discount factor and ε is the exploration factor. Source data for this table are provided as a Source Data file.

and actions in the task, rather than truly modelling the learning curve. Otherwise, the apparent superiority of RPE-biased replay may result from anomalous irregularities in the learning patterns and not true cognitive processes. Therefore, the same algorithms were trained on shuffled behavioural data in which the order of trials was randomly permuted 1000-fold. This preserved the average frequency of state-action pairs and their associated rewards, as well as the lengths of training sessions, but altered the learning curve including revaluation and reversal learning.

Overall, the errors for Q-learning with no replay were lower for shuffled data than real data, because shuffled behaviour was necessarily more consistent over time and therefore more predictable. Similarly to real data, error decreased sharply in early training sessions before reaching an asymptotic level (Fig. 6), because $Q$ values in early training sessions were distorted by unrepresentative rewards as a result of a small sample size of trials experienced. Unlike real data, the approach to asymptotic error was smooth and nearly monotonic.

Crucially, compared to the no-replay baseline, none of the replay policies improved error scores. This confirms that the improvement in error in the real data is a result of better predictions of the learning process, and not better convergence to general statistics in the task.

### Replay-biased RPE was the best predictor for all state-action pairs

We next accounted for the skew in training data towards the state-action pairs that were chosen most frequently. The transition from the high-probability arm to the mid-probability arm and vice versa (as they were in the initial and revaluation learning stages) were the most commonly experienced state-action pairs, representing 42% of trials overall, and the error was weighted by the frequency of each state such that errors in the more common states contributed more to the overall error than errors in the less common states. We therefore confirmed

**a**

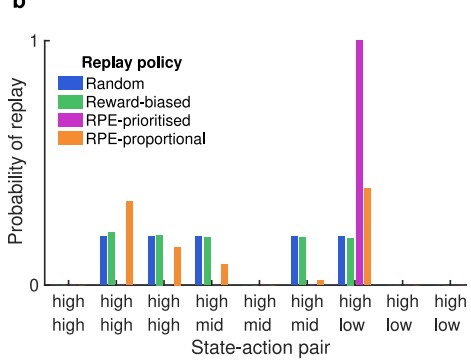

| Trial | State | Action | Reward | RPE   |
|-------|-------|--------|--------|-------|
| 1     | High  | Low    | 0      | -0.20 |
| 2     | Low   | High   | 1      | 0.73  |
| 3     | High  | Mid    | 1      | 0.82  |
| 4     | Mid   | High   | 1      | 0.71  |
| 5     | High  | Low    | 1      | 0.81  |
| 6     | Low   | High   | 0      | -0.27 |
| 7     | High  | Low    | 0      | -0.20 |
| 8     | Low   | Mid    | 0      | -0.18 |
| 9     | Mid   | High   | 1      | 0.70  |
| 10    | High  | Low    | 0      | -0.20 |

**b**

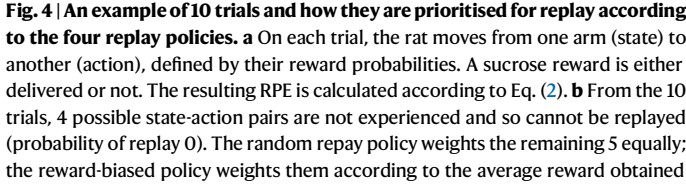

**c**

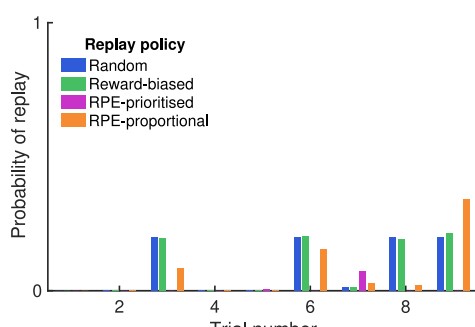

**Fig. 4 | An example of 10 trials and how they are prioritised for replay according to the four replay policies. a** On each trial, the rat moves from one arm (state) to another (action), defined by their reward probabilities. A sucrose reward is either delivered or not. The resulting RPE is calculated according to Eq. (2). **b** From the 10 trials, 4 possible state-action pairs are not experienced and so cannot be replayed (probability of replay 0). The random replay policy weights the remaining 5 equally; the reward-biased policy weights them according to the average reward obtained on trials corresponding to the state-action pair; the RPE-prioritised policy always replays the pair with the highest mean absolute recent RPE; and the RPE-proportional policy weights them in proportion to the mean absolute recent RPE. **c** After probabilistically selecting a state-action pair to replay (**b**), all replay policies select a trial corresponding to the pair with a recency bias. Source data for (**a**–**c**) are provided as a Source Data file.

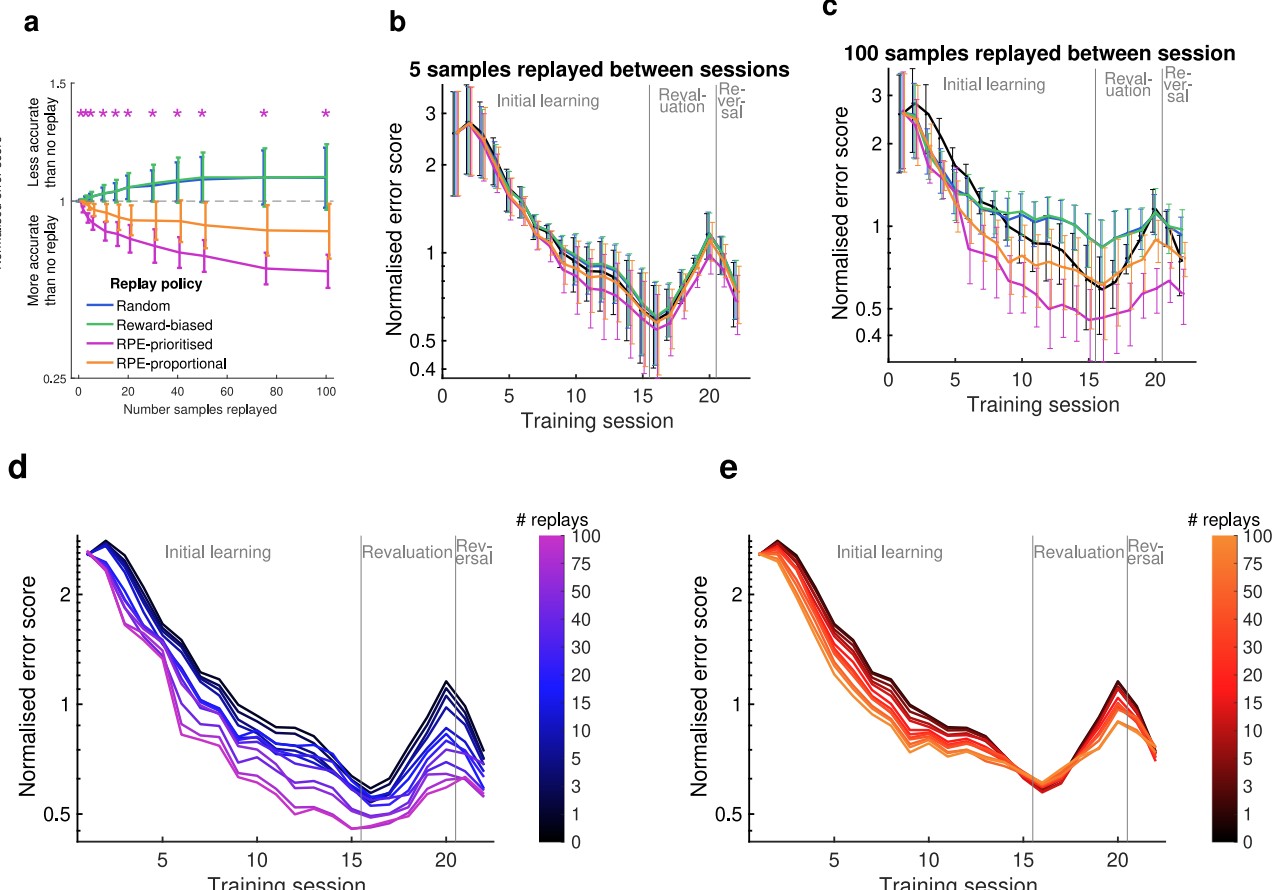

**Fig. 5 | Prediction accuracy of Q-learning model with four alternative replay policies. a** Normalised error score for each of 6 rats, with varying numbers of samples replayed between sessions, averaged over all trials for each rat, according to the four replay policies shown. Error scores normalised to the average error with no replay. Dashed line represents baseline with no replay. Error bars represent s.e.m. * indicates score for RPE-prioritised replay statistically different from 1 (one-sided linear mixed-effects model, $p < 0.05$, uncorrected). **b**, **c** Average error for each session, normalised to the average error for no-replay for each animal. With 1 sample replayed between each session (**b**) and 20 samples replayed between each session (**c**). Error bars represent s.e.m. **d**, **e** Average normalised error for each session, with varying numbers of samples replayed. **d** RPE-prioritised replay policy. **e** RPE-proportional replay policy. Source data for (**a**–**e**) are provided as a Source Data file.

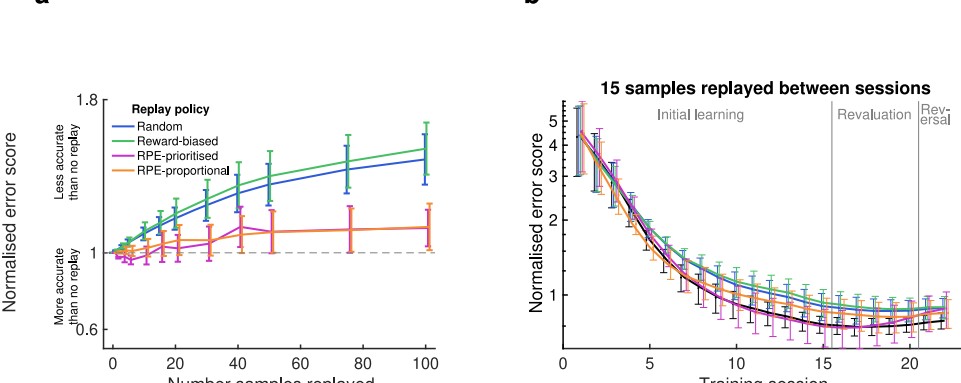

**Fig. 6 | Prediction accuracy of Q-learning model with five alternative replay policies on shuffled data. a** Normalised error score for each of 6 rats, with varying numbers of samples replayed between sessions, trained on shuffled data in which trial data (state, action and reward) are randomly permuted. Dashed line represents baseline with no replay. **b** Average error score for each session of shuffled data, normalised to the average error for no-replay for each animal, with 15 samples replayed between each session. Error bars represent s.e.m. Source data for (**a**, **b**) are provided as a Source Data file.

that Q-learning with RPE-biased replay learned to correctly predict all actions and not just the more-frequently chosen actions to which the cost function was skewed.

Figure S3 shows the improvement in error scores for each replay policy over no-replay baseline, for each state-action pair separately. Despite the skew in training data, the RPE-biased replay policies outperformed random and reward-biased replay policies for every state-action pair, although the improvement was not identical in each case. Nevertheless, the broad conclusion can be reached that RPE-biased replay policies better predicted

learning than either no-replay, random replay or reward-biased replay for all state-action pairs.

## A subpopulation of ventral striatal units encodes reward information

RPE signals have been hypothesised to be generated by the hippo-campus−striatal−VTA dopaminergic circuit, in which states are enco-ded by the hippocampus, reward predictions are generated in the ventral striatum, and RPE signals are computed by the VTA and broadcast back to the hippocampus and neocortex, potentiating synapses and offering a mechanism by which RPE might influence plasticity and learning[49–52]. The results of the modelling suggest that replay between sessions is influenced by such RPE signals, and should be observable in the single-unit activity in this circuit during post-task rest.

To test this, a separate cohort of three rats was trained on the same task for 17–20 sessions each, and implanted with silicon probes in both dorsal CA1 and ventral striatum enabling recording of extra-cellular unit activity during learning and for pre- and post-task rest periods. Rats underwent 12–15 sessions of an initial learning stage with reward probabilities of 87.5%, 50% and 12.5% on high-, medium- and low-probability arms respectively, followed by 5 sessions of reversal learning stage in which the reward probability of the high and low arms was swapped. Rats reached a greater-than-chance rate of optimal arm selection by day 5. A total of 617 CA1 units and 1406 striatal units were recorded, after excluding those with low isolation distance, and those from sessions where video tracking data of the animal's movement was unsuccessful.

Cells in the ventral striatum have previously been reported to encode many elements of behaviour, including upcoming action choice, predicted action outcome, current action, reward and RPE[53]. To compare with previous studies, striatal cells were divided into reward-modulated and non-reward-modulated by combining all trials in a given session and assessing whether firing rate varied significantly in 250 ms bins from the period −1 to +1 s around arrival at the reward location, compared to control time bins. A subset of striatal units, 232 of 1406 (17%) of the total, or 12.7–29.8% per rat, were categorised as reward-modulated according to this metric, similar to values reported previously (e.g. ref. [54]).

Trials typically consisted of two self-initiated runs separated by an imposed 5-s delay period: first towards the central platform, and sec-ond from the central platform to the reward location (Fig. 7a). Popu-lation activity in both CA1 (Fig. 7b) and ventral striatum (Fig. 7c) increased on approach to the reward location more markedly than on the approach to the central platform, indicating that activity in both areas was modulated by anticipation or prediction of immediate reward, not simply reflecting running behaviour. This is consistent with previous findings of ramping increases in ventral striatal firing rate on the approach to expected reward[55].

## Significant reactivation of intra-region and inter-region unit pairs in post-task rest

Previous studies have found significant reactivation of correlated activity in spatial tasks during post-task rest, both within the ventral striatum and between hippocampus and ventral striatum[25,54,56,57]. To confirm whether there was significant reactivation during post-task rest in these experiments, correlations between cell pairs were asses-sed during the TASK, PRE-task sharp-wave ripple periods and POST-task sharp-wave ripple periods to calculate the percentage of variance in POST correlations that could be explained by RUN correlations, controlling for PRE correlations. This approach was based on the explained variance (EV) metric also used by ref. [54] for hippocampal-striatal cell pairs, and[21] for other hippocampal-subcortical reactivation. Pooling across all 45 sessions from all rats, for pairs of CA1-CA1 cells there was an overall average EV of 0.24 and reverse explained variance

(REV) of 0.17 ($t(42) = 1.79$, $p = 0.0400$, one-sided paired $t$-test). EV and REV values were 0.32 and 0.10 ($t(23) = 6.33$, $p < 0.0001$, one-sided paired $t$-test) for striatal-striatal cell pairs, and 0.09 and 0.04 ($t(41) = 2.84$, $p = 0.0035$, one-sided paired $t$-test) for CA1-striatal cell pairs (Fig. 7d). Therefore CA1-CA1, striatal-striatal and CA1-striatial cell pairs showed significantly larger EV values compared to REV values, indicating TASK-dependent patterns of coactivity during POST, i.e. reactivation, both within and between brain regions.

## Reactivated cell pairs encode reward prediction

To interrogate the behavioural salience of the task-dependent reacti-vation implied by the EV analysis, we assessed the contributions of individual cell pairs and their behavioural correlates (see 'Methods').

We restricted the analysis to sessions in the initial learning stage when performance was significantly above 33% chance rate: at this level of performance, rats had acquired an association of higher reward probability or value to the high-probability arm than the medium-probability arm, which we refer to as reward prediction. CA1-striatal cell pairs were ranked according to their drop-one-cell-pair-out contribution to the session's EV-REV reactivation metric (see 'Meth-ods'; c.f.[21]), and the cell pairs with contributions in the highest decile and firing rate correlations higher during POST than PRE were labelled as reactivated cell pairs. Cell pairs with contributions in the smallest decile were used as a control population. Among 163 cell pairs classi-fied as reactivated, 52 (31.9%) comprised a reward-modulated striatal cell, compared to 50 out of 360 (13.9%) of non-reactivated cell-pairs, indicating a preference for reactivation of reward-related information between hippocampus and ventral striatum ($\chi^2(1) = 23.2$, $p < 0.0001$, $\chi^2$ test), consistent with previous observations[18,54].

We used the times during the TASK period when these cell pairs were coactive to indicate the behavioural correlates of the reactiva-tion: for each cell pair, the binwise minumum of their firing rates was calculated to create a measure of their coactivity (Fig. 7e–g). The z-scored coactivity averaged across medium-reward-expectation trials (both rewarded and unrewarded) showed a ramping up towards the point of arrival at the reward location that was stronger in the reacti-vated cell pairs than the control cell pairs (Fig. 8a). Z-scored coactivity averaged across high-reward-expectation trials showed a similar pat-tern, but with a higher peak just before arrival. A mixed-effects ANOVA comparing the peak coactivity for 163 reactivated versus 360 control cell pairs on high- versus medium-expectation arms showed a sig-nificant interaction effect between cell-pair type and trial type ($F(1) = 12.6$, $p = 0.0004$, two-sided; 8b). This effect was in addition to significantly greater coactivity of reactivated cell pairs for each trial type individually ($F(753) > 2.4$, $p < 0.0001$, post-hoc two-sided $t$-tests; Fig. 8b). A similar pattern was found for coactivity on rewarded trials only (Fig. S4). Thus, pairs of CA1 and ventral striatal cells displaying a higher degree of reactivation in post-task rest appear to be involved in encoding the anticipation of reward, and its expected probability, rather than reward outcome or error.

We then performed the same analysis for within-striatum reacti-vation: pairs of striatal-striatal cells were divided into reactivated and non-reactivated according to their contribution to the overall EV-REV metric for within-striatum reactivation. On rewarded trials, the reac-tivated pairs' z-scored coactivity showed a similar ramp up in antici-pation of reward, plus a subsequent increase in coactivity in the 5 s following reward delivery on the medium-reward-expectation arm (i.e. corresponding to high, positive reward-prediction error) that was not present in the 5 s following reward delivery on the high-reward-expectation arm (i.e. corresponding to low, positive reward-predic-tion-error; Fig. 8c). This was confirmed by a mixed-effects ANOVA comparing the peak coactivity for reactivated versus control cell pairs in the 5 s following reward delivery on high- versus medium-expectation rewards, which shows a significant interaction effect between cell-pair type and trial type ($F(1) = 8.6$, $p = 0.0035$, two-sided;

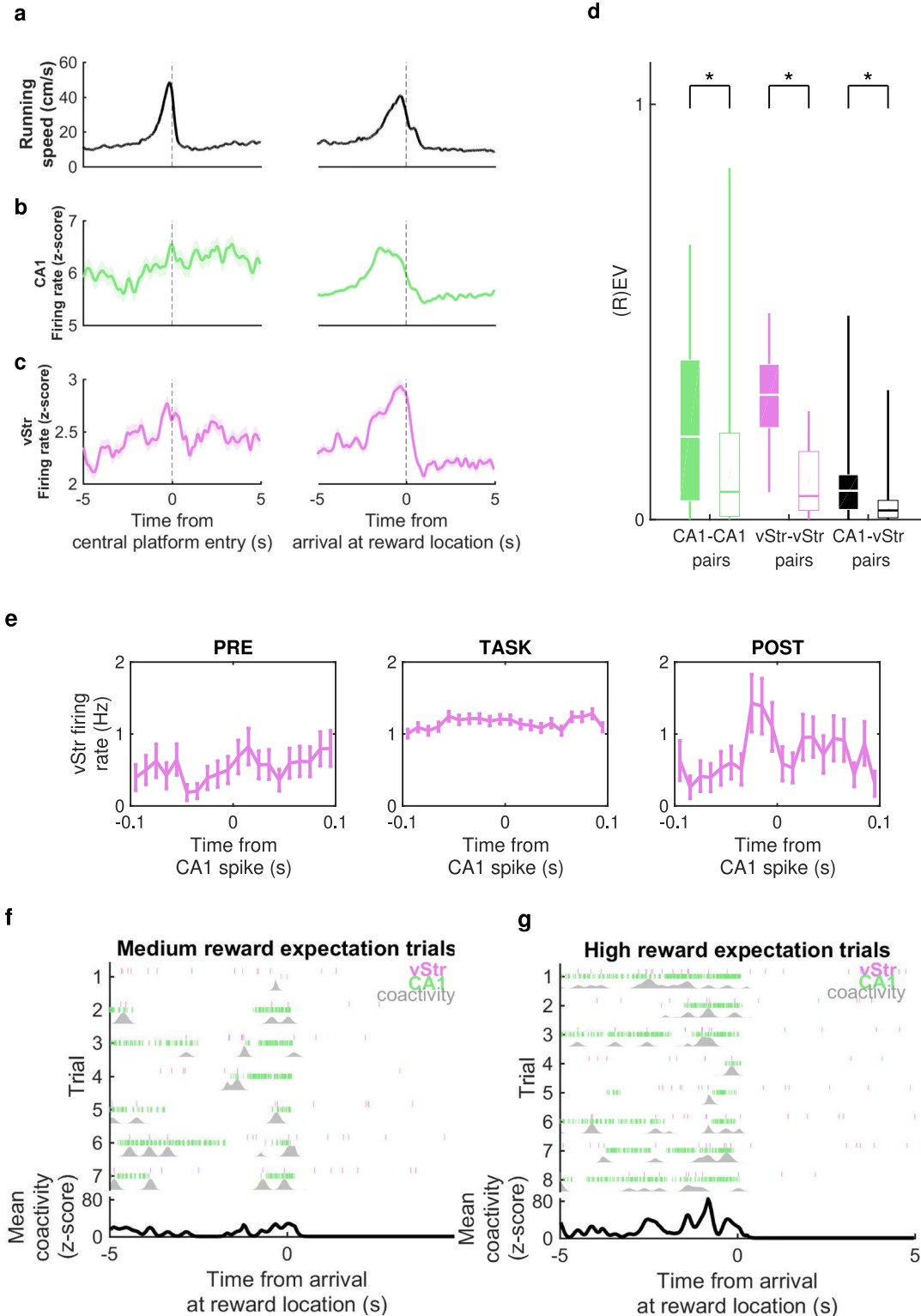

Fig. 8d). In contrast to pairs of CA1-ventral-striatal cells' reactivation of reward-prediction signals, striatal-striatal cell pairs therefore showed preferential reactivation of RPE signals.

## Discussion

We trained rats on a reinforcement learning task designed to dissociate reward outcome (presence or absence of reward) from reward prediction error (RPE; an unexpected reward or absence of reward) on each trial. Training variations of a Q-learning reinforcement learning model to predict behaviour on the task revealed that Q-learning with

replay prioritised by RPE was the best predictor of learning. Consistent with this, we found that pairs of CA1-ventral striatal cells which are the most strongly reactivated during post-task rest encode reward prediction, ramping up to the point of reward delivery, while pairs of ventral striatal cells encode RPEs, being more strongly coactivated following less certain reward.

Our first main result was that Q-learning can model rats' learning of the stochastic reinforcement learning task, producing low reliability-errors when trained on rats' behaviour and predicting the likelihood of actions on each trial. This is consistent with other studies

**Fig. 7 | Electrophysiological data from CA1 and ventral striatum (vStr).**
**a** Mean ± standard error (s.e.m.) running speed around the two main events of each trial: entry to the central platform between the three arms (left), and subsequent arrival at the reward location on the chosen arm (right); all recording sessions pooled. **b** Mean ± s.e.m. firing rate of CA1 cells around the same two events. **c** Mean ± s.e.m. firing rate of ventral striatum cells around the same two events. **d** Explained variance (EV; filled bars) and reverse explained variance (REV; open bars) for intra- and inter-regional cell pairs during concatenated ripple activity in 2 h of PRE- and POST-task rest. Whiskers represent range, boxes represent interquartile range, centres indicate median. * indicates significant difference between EV and REV (one-sided paired *t*-test, *p* = 0.04 for CA1-CA1 pairs from *n* = 45 sessions, *p* = 9*e*⁻⁷ for vStr-vStr pairs from *n* = 25 sessions, *p* = 0.004 for CA1-vStr pairs from

*n* = 44 sessions, uncorrected). **e** An example reactivated pair of CA1-vStr cells which contributed highly to the session's EV-REV value: spike-triggered average firing rate of the ventral striatum cell around CA1 cell spikes, during ripples in PRE and POST and for the whole TASK epochs, at 10 ms bins, and error bars showing s.e.m. **f** Event-triggered activity of the same reactivated cell pair in e: pink ticks show timing of spikes of the vStr cell and green ticks show timing of the CA1 cell over all arrivals at the reward location where a medium reward probability is expected. Grey shows the coactivity, i.e. the minimum firing rate between the two. Lower black trace shows the mean coactivity over trials, z-scored relative to the whole recording session. **g** As (**f**), for the same reactivated cell pair, for trial where a high reward probability is expected. Source data for (**a**–**g**) are provided as a Source Data file.

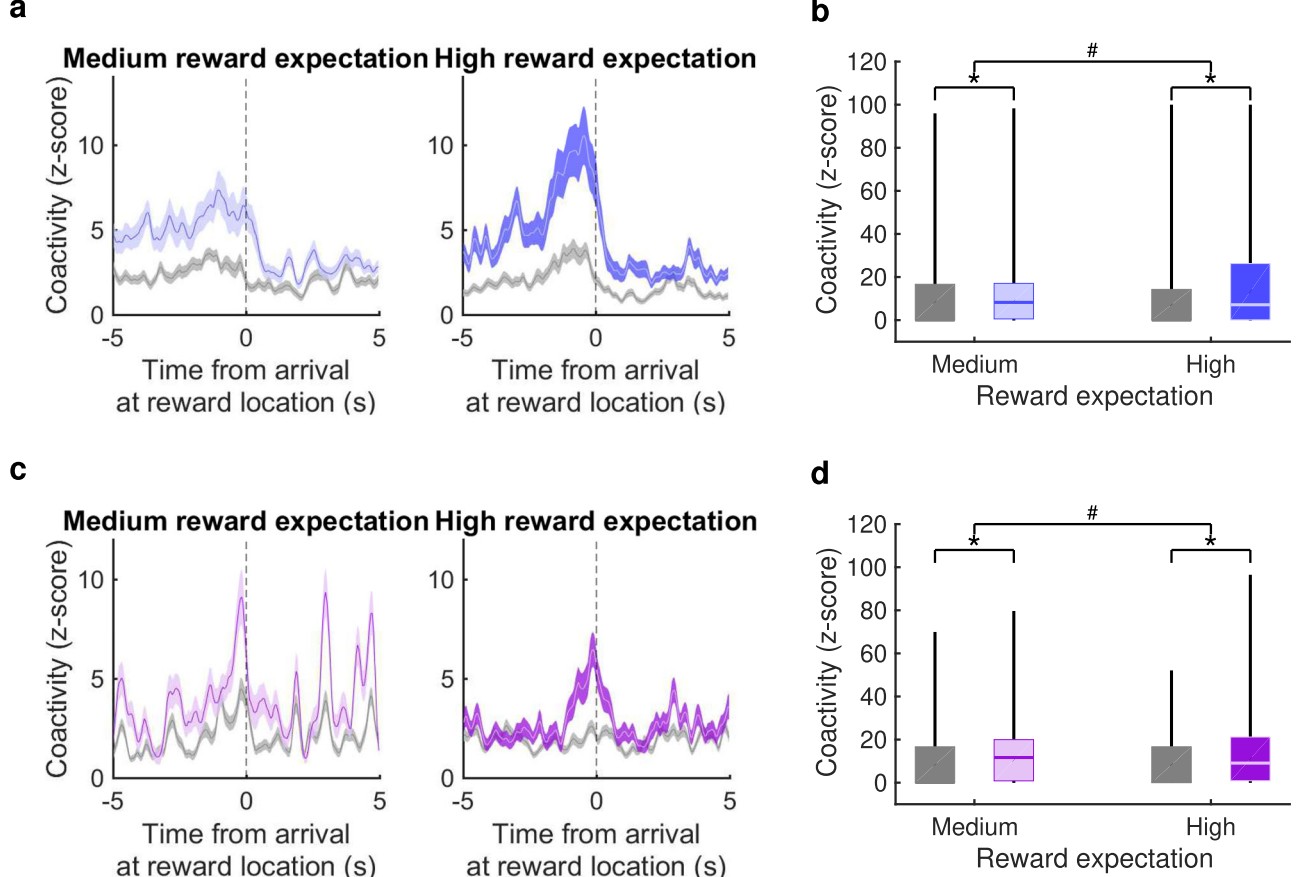

**Fig. 8 | Coactivity of CA1-striatal cell pairs around the time of approach to reward location. a** Mean ± s.e.m. z-scored coactivity of reactivated CA1-vStr cell pairs (blue) and non-reactivated CA1-vStr cell pairs (grey) around the time of arrival at reward locations on medium- and high-expected reward trials. **b** Average coactivity in the 2 s prior to arrival at the reward location (shown in h) for 163 reactivated cell pairs (blue) and 360 non-reactivated cell pairs (grey). Whiskers represent range, boxes represent interquartile range, centres indicate median. Asterisks for medium and high respectively indicate statistical significance between reactivated and non-

reactivated pairs (post-hoc two-sided *t*-tests, *p* = 4*e*⁻⁵ for Medium, *p* = 6*e*⁻⁷ for High, uncorrected); hash between medium and high indicates statistical significance of the interaction effect between reward expectation and cell-pair type (*p* = 4*e*⁻⁴, uncorrected). **c** As a, for vStr-vStr cell pairs, reactivated (purple) and non-reactivated (grey), on rewarded trials. **d** As (**b**), for coactivity of 204 reactivated and 551 non-reactivated vStr-vStr cell pairs in the 5 s after arrival at the reward location, when reward was delivered (*p* = 4*e*⁻⁵ for Medium, *p* = 0.01 for High, *p* = 0.004 for interaction). Source data for (**a**–**d**) are provided as a Source Data file.

showing that Q-learning can predict behaviour in a range of tasks in rodents monkeys and humans[34]. Given this result, we then hypothesised that adding replay to the Q-learning model between sessions might better reflect learning and therefore better predict behaviour. However, a policy of replaying state-action pairs randomly did not produce lower errors overall, indicating a poor model of the cognitive processes underlying reinforcement learning. Similarly, biasing replay by sampling from state-action pairs which had produced the largest recent reward did not produce lower errors relative to no-replay.

In contrast, biasing replay by sampling from state-action pairs which had produced the largest recent RPE decreased reliability

errors, demonstrating that the cognitive processes involved in the learning of this task are influenced by offline activity that takes place between sessions biased by RPE. This result did not hold when training data was shuffled, demonstrating that the influence of RPE is a feature of the learning process and not an epiphenomenon resulting from the general statistics of behaviour. Moreover, the result did hold for all state-action pairs, despite the over-representation in training data of those most frequently experienced. This gives credence to the notion that the Q-learning model with replay biased by RPE is a good overall model of state-action values held by the brain and offers a viable means to extend

hippocampus-based models of replay's contributions to spatial memory[58].

Performance on memory tasks has widely been found to improve following a period of sleep[59–61], associated with replay of activity which codes recent experiences during hippocampal sharp-wave ripples[3]. Associations between spatial location and reward or action values are encoded in the ventral striatum, which receives direct inputs from dorsal CA1 whose activation after learning is required to consolidate spatial memories[62,63]. The modelling results predict post-task reactivation of such connectivity within the hippocampal-striatal network to induce long-term potentiation at the synapses active during replay. Accordingly, we found reactivation in hippocampal-striatal cell pairs, with an increase in cell-pair coactivation particularly for cell pairs whose coactivity was higher on the approach to high-probability rewards than medium-probability rewards. We also found reactivation in striatal-striatal cell pairs, with an increase in coactivation for pairs whose activity was higher following less-expected reward than more-expected reward. These represent a reward-prediction signal and reward-prediction-error signal, respectively, consistent with Q-learning, supporting the hypothesis that hippocampal replay modulates the midbrain circuit responsible for updating reward predictions and RPEs. The reactivated hippocampal-striatal cell pairs showed a ramping pattern on the approach to reward location, which has been shown to reflect a dopaminergic RPE signal. While various studies report projections from hippocampus to ventral striatum, there are no known projections from ventral striatum to hippocampus[64], which implies that this coactivation during learning and reactivation during post-task rest are both driven by the hippocampus, perhaps as part of a broader network incorporating other brain areas including VTA and prefrontal cortex. Being limited to these particular recording areas gives a narrow view of the possible physiological implementations of the modelling results, and cannot serve as direct tests of the competing hypotheses which could rely on unobserved parts of the circuit. We therefore propose that post-task replay underlies the RPE-biased offline updating of state-action values which influenced reinforcement learning in this task.

The apparent dual computational function of reactivation between and within brain areas likely reflects the distributed nature of reinforcement learning in the hippocampal-striatal-VTA circuit. Similar simultaneous but distinct replay patterns have been observed between the hippocampus and entorhinal cortex[65], and between hippocampus and prefrontal cortex[66]. Further investigation of how hippocampal-hippocampal, hippocampal-striatal and striatal-striatal replay events are temporally or computationally related would be valuable for elucidating how offline activity influences learning processes. One interpretation of the electrophysiological results here is that hippocampal-striatal reactivation is biased by reward prediction to reinforce the learned $Q$ values, while striatal-striatal reactivation is biased by RPEs to update the $Q$ values. Another interpretation is that striatal-striatal reactivation follows the RPE-biased sample selection predicted by our modelling, while hippocampal-striatal reactivation follows a policy-biased (replaying the most likely upcoming paths)[67] or experience-biased (replaying the most frequently experienced paths)[24] sample selection.

The suggestion that hippocampal replay might be biased by RPEs differs from the commonly held view that replay is biased by reward itself[4,19,20,68–70]. However, the studies on which this conclusion is based generally do not use tasks which explicitly dissociate reward from RPE, so these results in the literature are not inconsistent with our suggestion that RPE biases replay.

Despite the prevalence of the idea that reward biases replay, our alternative theory that RPE biases replay fits better with existing research into the roles of dopamine. Dopaminergic projections from the VTA to CA1 in the hippocampus have been found to modulate both replay during sleep following exposure to a novel environment, and

subsequent memory performance in the same environment[71]. It is suggested that dopaminergic neuromodulation might tag synapses by upregulating plasticity-related proteins, causing long-lasting potentiation which allows the stabilisation of the memory trace during subsequent sleep and rest[72,73]. Phasic dopaminergic inputs to the hippocampus are triggered not only in response to novelty, but also in the context of reward[49], offering a likely mechanism by which reward-related information might influence replay. Indeed, replay has been found in reward-related VTA cells[74,75], confirming the involvement of the full hippocampal-striatal-VTA loop in post-task reactivation.

Several studies have expressly linked replay to reward, ostensibly in contrast with our results, but often RPE is a confounding factor in these which cannot be discounted. In humans, high monetary reward (but not low monetary reward) is linked to sleep-dependent improvements in associative memory[76,77]; in these human studies, RPE was not estimated but would presumably be higher overall in the high-reward than low-reward condition, conflating reward-dependent effects with RPE-dependent effects. In rodents, newly-rewarded behaviour has been associated with replay more than behaviour which had been rewarded in previous sessions[19]; the authors attributed this replay bias to novelty, but it is also consistent with increased RPE when new behaviours are rewarded for the first time. Moreover, following extended reinforcement of both behaviours, the replay bias for the newly-rewarded behaviour was eliminated. In a third study, results were more mixed: following an increase in reward magnitude at one end of a linear track, there was more replay associated with the larger-magnitude end than the unchanged-magnitude end, correlated with both reward and RPE[68]. However, following an elimination of reward at one end, there was a reduction in replay following a reduction in reward despite the increase in RPE. This is more consistent with reward-biased than RPE-biased replay, although the authors noted a rebound effect when the eliminated reward was reinstated: greater replay was found at the reinstated-reward end than the unchanged-reward end, despite identical reward magnitudes. This leaves open the possibility of bias by positive over negative RPEs. A fourth study found more replay of large-reward-related activity than small-reward-related activity on a maze task[16], but because reward was received on every trial analysed, any effects of reward magnitude are conflated with positive RPE.

Conversely, the specific case for RPE-biased replay is supported by findings that neural sensitivity to RPEs in humans predicts the amount of awake replay during a reinforcement learning task, and replay amount correlated with subsequent performance in a task requiring behavioural flexibility[78].

In addition to human and rodent studies, findings from the literature on machine learning show some consistency with our results. A number of machine learning studies have found that storing new information in memory buffers and sampling from it at regular intervals, similar to hippocampal replay, can speed up learning[47,79–81] and more so when replay is biased by prediction errors[82,83]. RPE-biased replay may therefore represent an adaptive focus whereby resources are focused on areas of a cognitive model which needs updating[84–86].

We do not claim that this tells the whole story: RPE is highly unlikely to be the only factor that biases replay and the phenomenon is likely to be much more multifaceted than this model suggests. First, phasic dopamine signalling to hippocampus may encode other kinds of prediction errors or aspects of reward to which the VTA is sensitive[87–92], and bias replay by the same mechanism. Reward itself may bias replay, especially if positive RPEs influence replay more than negative RPEs; there is also evidence that novelty[93,94], the expectation of reward[70], frequency of experience[95] and strength of encoding[96] bias replay too. Furthermore, in addition to aiding reinforcement learning, replay has been associated with other memory-related functions including planning[5,97], processing of emotional memories[98], creative problem-solving[99] and generalising from episodic memories to

abstractions[7,100], all of which are likely to necessitate some biasing of replay distinct from RPEs. In sum, while we fully expect replay to be more complex, we have focused on one facet with important neuro-biological foundations.

Our model assumes that a cache of all experience is stored from which to be sampled, which is expensive and unrealistic at large scales. This may not be necessary if memory for individual trials is gradually forgotten and subsumed into cortical long-term memory, for example over the course of hours over which cell assembly activation decays[101].

Finally, this model leaves open some questions. Although the role of post-task VTA activity in influencing future reward-related behaviour has been demonstrated previously[75,102], it remains unclear how this part of the hippocampal-striatal-VTA loop contributes to replay in this task. There is also an open question about possible diverging roles of replay during behaviour compared to prolonged rest and sleep. Here, we have considered replay between sessions, which is likely to take place at least partly during sleep; but replay during wake has also been shown to be necessary for learning[15].

In summary, we found that a Q-learning-based reinforcement learning model which assumes offline updates between sessions is a better predictor of learning behaviour than one which does not assume offline updates. Specifically, this is true when updates are prioritised according to experiences that have recently elicited high RPEs, and not when they are prioritised according to reward or random recent experiences. Activity reflecting reward-prediction signals in the CA1-ventral-striatal network and RPEs in the striatal network is reactivated, demonstrating a mechanism by which state-action values across hippocampus and striatum may be updated offline. This finding offers a refined interpretation of how offline activity during rest and sleep might aid reinforcement learning, in terms of RPE rather than solely reward.

## Methods

### Behavioural task

All procedures were performed in accordance with the United Kingdom Animals (Scientific Procedures) Act 1986 and European Union Directive 2010/63/EU and were reviewed by the University of Bristol Animal Welfare and Ethical Review Board.

Six adult male Lister hooded rats in the first cohort (weighing 260-330g) and three adult male Lister hooded rats in the second cohort (weighing 300–430 g, Charles River Laboratories, UK) were individually housed with environmental enrichment, and food-restricted to no less than 85% of their pre-restriction body weight. Following habituation to the recording room, they were trained during the light part of a 12:12 light/dark cycle to forage on a 3-armed radial maze for liquid sucrose rewards in a dimly-lit room. The maze consisted of a raised central platform 25cm in diameter, with three arms (60 cm × 7 cm) protruding from it (Fig. 1a). Arms were separated from the central platform by inverted-guillotine pneumatic doors, which raised to block access to the arms, and fell below the maze floor to allow access. Turning zones (10 cm × 10 cm) with lick ports were positioned at the end of each arm, at which 20% sucrose solution rewards were delivered. Door movements and reward delivery were operated automatically according to the animal's position, tracked using a webcam mounted above the maze, using custom MATLAB (The MathWorks) code. Following at least three days of habituation to the recording room and maze-operation sounds, each animal performed 17–22 once-daily training sessions, between 5 and 7 days per week, lasting 1 h each.

Trials began when a rat entered, or was placed by the experimenter on, the central platform with all doors closed. Doors opened following a 5-s delay period. When the animal reached the lick port, reward was probabilistically delivered or withheld, and doors to the other two arms were closed; the third door was closed when the animal re-entered the central platform to begin a new trial.

Each arm was assigned as either high probability, mid probability or low probability, which determined the protocol for reward delivery. These assignments remained fixed throughout training for each animal, but were counter-balanced between animals. The cohort of rats on which the behavioural model was fit underwent three learning stages with three sets of reward probabilities. In the initial learning stage, sessions 1–15, the high-probability arm delivered a reward on 6 out of 8 (75%) legitimate entries to the arm, the mid-probability arm on 4 out of 8 (50%), and the low-probability arm on 2 out of 8 (25%). A legitimate entry was one in which a different arm had been entered on the previous trial; entering the same arm twice in a row was incorrect and did not result in a reward delivery. In the revaluation stage, sessions 16–20, the reward probabilities for the high- and low-probability arms were amplified: reward was delivered on 7 out of 8 (87.5%) and 1 out of 8 (12.5%) legitimate entries respectively. In the reversal learning stage, sessions 21–22, the reward probabilities for the high- and low-probability arms were switched, such that the (formerly) high- and low-probability arms delivered reward on 1 out of 8 (12.5%) and 7 out of 8 (87.5%) of legitimate entries respectively.

The cohort of rats from which hippocampal and striatal activity was recorded underwent just one change in reward probabilities. In the first 12-15 sessions, the high-probability arm delivered a reward on 7 out of 8 (87.5%) legitimate entries to the arm, the mid-probability arm on 4 out of 8 (50%), and the low-probability arm on 1 out of 8 (12.5%). In the remaining 5 sessions, the reward probabilities for the high- and low-probability arms were switched.

For this cohort, training sessions were flanked by rest sessions in the home cage of ~2 h before and after training.

### Q-learning

We trained several variations of a Q-learning algorithm on the behavioural data to predict choices of which arm would be entered on each trial. Q-learning is a reinforcement learning algorithm developed for Markov decision processes in which an agent selects actions in its environment and observes the outcome, recording at each time step $t$ its starting state $s_t$, selected action $a_t$, resulting reward $r_t$ and resulting state $s_{t+1}$. The agent builds up a matrix $Q$ of $Q$ value estimates for every state-action pair (1) corresponding to the future discounted expected reward, i.e. the temporal difference between the current state and the reward state. These $Q$ value estimates are used to guide actions to maximise reward. At each time step $t$, the $Q$ value for the state-action pair observed is updated by (2), where $\alpha \in (0, 1)$ is a learning rate parameter which determines the degree to which new information overrides old information, and $\gamma \in (0, 1)$ is a discount parameter which determines the importance of long-term gains.

In this task, entries into a chosen arm (and arrival at the goal location at the end of the arm) were modelled as actions, while the arm entered on the previous trial, on which reward probabilities were contingent, were modelled as states. Each trial therefore gave rise to one state-action transition out of nine possible state-action pairs. Actions were selected according to probabilities $p_a$ for each action $a$, determined by $Q$ values and an exploration-exploitation parameter epsilon:

$$p_a = \frac{e^{\epsilon Q_{s,a}}}{\sum_{a=1}^{3} e^{\epsilon Q_{s,a}}} \tag{3}$$

To reflect rats' natural tendency to alternate between options, $Q$ values were initialised before learning to:

$$\begin{bmatrix} 0 & 0.7 & 0.7 \\ 0.7 & 0 & 0.7 \\ 0.7 & 0.7 & 0 \end{bmatrix} \tag{4}$$

## Q-learning with replay

We used four variants of Q-learning in which additional nominal offline updates are performed between performed online trials, based on sequences already experienced, to boost learning. This has the effect of learning from several trials per actual trial of experience, and is similar to the Dyna-Q algorithm which has been shown to speed up learning compared to Q-learning alone[103] in a manner which may underlie the function of hippocampal replay[44]. Generally, sequences are selected randomly from a memory buffer of recently-acquired experiences, without bias towards any trial or type of trial. Given the observed bias reported in the literature towards salient experiences, such as those rewarded or aversive, we modified Dyna-Q to perform updates only between sessions and to reflect hypothesised biases in four different ways.

## Parameter-fitting

**Parameter-fitting for Q-learning.** First, a Q-learning algorithm (without replay) was trained, to obtain a baseline score against which various replay policies could be compared. $Q$ values were stored for each state-action pair on the task, and updated according to each animal's experience. A state $s_t$ was defined as the arm visited on the previous trial $t-1$, and an action $a_t$ was defined as the arm chosen on the current trial $t$. Following each trial of an animal's training, the $Q$ value $Q(s_t, a_t)$ was updated according to the reward received, $r \in \{0, 1\}$ by equation (2), and $Q$ values were transformed into a forecast probability of choosing each arm on the subsequent trial.

The learning rate $\alpha$, discount factor $\gamma$, and exploration factor $\epsilon$ were free parameters that were tuned to each rat, using the following optimisation procedure. Here we used an error score adapted from the reliability component of ref. 104 and generated based on the forecast probabilities of all trials, to quantify the consistency of the forecast probabilities with the animals' behaviour. The mean observed frequency was calculated for each state-action pair, i.e. the proportion of trials on which a given action was chosen in a given state, and the error score $R_t$ for a given trial $t$ was calculated according to:

$$R_t = n_{s_t} \cdot \sum_{a=1}^{n_a} (p_a - o_{s_t, a})^2 \tag{5}$$

where $s_t$ is the animal's state on trial $t$, $n_{s_t}$ is the number of trials on which the animal was in state $s_t$, $n_a$ is the number of possible actions (3) $p_a$ is the forecast probability for entering arm $a$, and $o_{s,a}$ is the mean observed frequency of state-action pair $s, a$.

Parameter optimisation was performed using Bayesian adaptive direct search (BADS)[105], with the error score averaged over 25 runs with different seeds used as the objective function to reduce its stochasticity. Analyses were performed on the average error over 1000 runs with seeds separate from those used during parameter optimisation, using the resulting parameter values.

**Parameter-fitting for Q-learning with replay.** Against the baseline of no-replay, the same optimisation procedure was performed with increasing amounts of replay according to four replay policies. Following each session, a specified number of samples were chosen from all the trials experienced so far. How the samples were selected depended on the replay policy (detailed below); a probability $P(s, a)$ was assigned to each state-action pair to determine which pair to sample from. From the chosen state-action pair, a sample trial was chosen according to the probability $P(i)$ in which a recency parameter ensured that more recent trials were exponentially more likely to be chosen. $Q$ values were then updated according to the state, action and reward of the sampled trial, in the same manner as so-called online $Q$ value updates described in equation (2).

Each replay policy required the same three parameters to be optimised as in Q-learning without replay, plus additional parameters

**Table 2 | Number of free parameters for each replay policy**

| Replay policy | Number of parameters |
|---|---|
| No replay | 3 |
| Random replay | 4 |
| Reward-biased replay | 4 |
| RPE-prioritised replay | 5 |
| RPE-proportional replay | 5 |

for recency and/or RPE-weighting. Table 2 shows the number of free parameters for each replay policy.

These were optimised according to the same procedure as for Q-learning with no replay, described above, for $n = \{1, 3, 5, 10, 15, 20, 30, 40, 50, 75, 100\}$ replay events between each session, resulting in 11 sets of parameter values for each replay policy and each animal. Comparing this to plausible quantities of replay events in animals is not trivial, but studies in which discrete replay events are enumerated report 100–200 bursts of hippocampal activity that can be statistically related to prior experience, over the first 1 or 2 h after experience[16,106]. Separately, reactivation of cell pairs has been found to decay to baseline well within that time period following exposure to familiar environments[101], so the first 1–2 h is likely to be when most replay of recent experience in a familiar environment occurs.

### Random replay

Random replay, biased by nothing but the recency of an action, was included as a control. For each replay event, a state-action pair was chosen at random out of all state-action pairs experienced so far:

$$P(s, a) = \frac{1}{n_{sa}} \tag{6}$$

where $n_{sa}$ is the number of state-action pairs experienced (up to 9). The subset of trials experienced, $i \in (1, I)$, which represented this state-action pair were ordered chronologically, and the probability $P(i)$ of a trial $i$ being replayed was determined according to a recency parameter $\varphi$:

$$P(i) = \frac{i^{\varphi}}{\sum_{j=1}^{I} j^{\varphi}} \tag{7}$$

### Reward-biased replay

Reward-biased replay represents the predominant interpretation of how reward influences replay[69,107]. For each replay event, a state-action pair $s, a$ was chosen probabilistically in proportion to its $Q$ value:

$$P(s, a) = \frac{Q(s, a)}{\sum_{s=1}^{n_s} \sum_{a=1}^{n_a} Q(s, a)} \tag{8}$$

The subset of trials experienced which represented the chosen state-action pair were ordered chronologically, and determined according to equation (7).

### RPE-prioritised replay

RPE-prioritised replay represents the policy of replaying trials associated with the most surprising outcomes, i.e. where the difference between expectation ($Q$ values) and experience (reward) was greatest. For each trial $t$, RPE was calculated as the difference $\delta$ between actual reward and expected reward:

$$\delta_t = r + \gamma \cdot Q(s_{t+1}, a') - Q(s_t, a_t) \tag{9}$$

where $a'$ is the action with the highest $Q$ value in state $s_{t+1}$.

For every trial $i \in (1, I)$ which was an example of a given state-action pair, its absolute value was weighted, determined by a

parameter $\psi$ raised to the power of its recency $i$:

$$\Delta_i = |\delta_i| \cdot \psi^i \tag{10}$$

The weighted RPEs, $\Delta$, were then averaged to produce an overall weighted-average RPE, $\overline{\Delta}_{s,a}$, for each state-action pair $s$, $a$, which was more heavily influenced by recent trials:

$$\overline{\Delta}_{s,a} = \frac{\sum_{i=1}^{I} \Delta_i}{I} \tag{11}$$

The state-action pair with the highest $\overline{\Delta}_{s,a}$ was selected, and the subset of trials experienced which represented the chosen pair were ordered chronologically, and determined according to equation (7). Once replayed, the $\delta_t$ for the trial sampled was updated to reflect the RPE resulting from the replay event.

### RPE-proportional replay

RPE-proportional replay is a variant of RPE-prioritised replay, in which state-action pairs are chosen in proportion to their weighted-average-RPE instead of choosing the pair with the highest weighted-average-RPE. The RPE was calculated according to eq. (11) and a state-action pair to be sampled from was chosen probabilistically according to:

$$p_{s,a} = \frac{\overline{\Delta}_{s,a}}{\sum \overline{\Delta}_{s,a}} \tag{12}$$

The subset of trials experienced which represented the chosen state-action pair were ordered chronologically, and determined according to equation (7). Once replayed, the $\delta_t$ for the trial sampled was updated to reflect the RPE resulting from the replay event.

**Shuffling procedure.** As an additional control, the parameters were also optimised for shuffled data, in which trial order was randomly permuted 1000-fold. This preserved the large-scale information in the training data, such as the mean observed frequency and average rewards of state-action pairs and the number of trials in each session between replays, but disrupted the specific structure of how this information was acquired over time.

### Electrophysiology

Three rats were implanted with a 9mm, 2-shank H2 silicon probe and a 9mm, 4-shank E silicon probe (Cambridge NeuroTech, UK), each with 64 recording sites, targeted at dorsal CA1 and ventral striatum, respectively. Probes were mounted on aluminium blocks (7.5 mm × 3.3 mm × 3.0 mm) and targeted at 2.1 mm lateral, 4mm posterior and 2.5 mm ventral to bregma (CA1) and 1.5 mm lateral, 1.7 mm anterior and 7 mm ventral to bregma (striatum), in the right hemisphere, based on the atlas of[108]. Surgery was performed under isoflurane recovery anaesthesia in sterile conditions and probes cemented to the skull using Gentamycin-impregnated bone cement (dePuy CMW). A subcutaneous injection of the analgesic buprenorphine (0.05 mg/kg) was given post-surgery.

Extracellular recordings were made using an Open Ephys acquisition system at a sampling rate of 30 kHz, with two RHD2164 headstages, one with an integrated accelerometer. Recordings were referenced to a stainless steel screw implanted over the cerebellum. A red LED was attached to the implant, and the session was recorded by a ceiling-mounted webcam which allowed the rat's movement to be tracked. Electrophysiological recordings and position tracking were synchronised post-hoc using a second LED which blinked at random intervals.

Raw data were automatically spike-sorted using Kilosort software[109] and manually curated using Phy (https://github.com/cortex-lab/phy). In brief, raw data were common-average referenced, high-pass filtered and whitened to remove correlated noise, before prototypical spikes were detected whenever the amplitude exceeded a given threshold. Detection and clustering of dimensionality-reduced spike waveforms were then optimised iteratively using a template-matching procedure. In the manual curation step, clusters were merged, accepted or rejected as noise by visual inspection, according to their inter-spike interval histograms, amplitude and spike waveform. Finally, clusters were restricted to those with an isolation distance of >15[110].

### Data analysis

**Reward-related firing.** Following[54], spike trains of ventral striatal cells were divided into 250 ms bins, centred around the time of arrival at reward location, and averaged across trials. A cell's mean firing rate in each of the 8 bins from −1 to +1 s was compared to firing during 3 control bins using Wilcoxon's signed rank test. Cells for which at least one bin was significantly different from all 3 control bins were classified as reward-responsive, using an alpha value of 0.05.

To analyse striatal cells' encoding of reward expectation, binless spike trains equivalent to 50 ms bins[111] were z-scored with respect to the whole training session. Analysis was restricted to sessions in the initial learning stage in which performance was above chance and before reward probabilities changed. Cells which showed a peak firing rate in the 2-s period before arrival at reward location, before the reward outcome (reward or no reward) was known, exceeding 2 standard deviations were classified as encoding reward expectation. The same 2-s period was compared for arrival at the high-probability reward location and the mid-probability reward location, pooled across rats, using a paired $t$-test to test for differences in population-level firing.

**Sharp-wave ripple detection.** Sharp-wave ripples were detected using the SleepWalker toolbox in MATLAB (https://gitlab.com/ubartsch/sleepwalker). Hippocampal LFP was filtered at 120–250 Hz, and events were extracted when ripple power exceeded 3.5 standard deviations above the mean, and no more than 25 standard deviations. Events with a duration of 10–500 ms, an amplitude of 30–1000 μV, and separated by at least 30 ms were included as ripples.

**Explained variance and reverse explained variance.** To analyse ripple-related reactivation, sessions with at least 5 CA1 and 5 ventral striatal cells were included. The PRE and POST periods were restricted to concatenated windows of 200 ms from each ripple peak. Pearson's correlation coefficients were calculated between binless spike trains equivalent to 50 ms bins in the PRE, TASK and POST periods separately and combined to create three correlation matrices. The similarity between PRE, TASK and POST was calculated by taking the correlation coefficient $r$ between their correlation matrices[94]:

$$EV = \left( \frac{r_{TASK,POST} - r_{TASK,PRE} r_{POST,PRE}}{\sqrt{(1 - r_{TASK,PRE}^2)(1 - r_{POST,PRE}^2)}} \right)^2 \tag{13}$$

giving a measure of the partial correlation between cell-pair coactivity during post-task ripples with that during the task, controlling for cell-pair coactivity during pre-task coactivity.

REV was calculated by exchanging $r_{PRE}$ and $r_{POST}$ in eq. (13).

**Experience-dependent increases in cell-pair coactivity during sleep and rest.** The contribution of each CA1-striatal or striatal-striatal cell pair to overall inter-region reactivation was measured by recalculating EV-REV with the cell pair removed and subtracted from the session's overall EV-REV value. A threshold of the top decile within each session was used to classify candidate reactivated cell pairs (the analysis was also repeated for the top 5% and the top 20% with similar results). Mathematically, EV-REV can be driven by cell pairs whose correlation gets stronger from PRE to TASK and stays strong in POST, or whose correlation weakens from PRE to TASK and stays low in POST.

The former could be said to carry or encode reactivated content, while the latter reflects more general network reorganisation without encoding task-relevant information. Therefore, from this top decile, only the cell pairs whose correlation increased from PRE to POST were included as reactivated cell pairs. These reactivated cell pairs were compared to the decile that had the lowest magnitude of contributions to EV-REV (i.e. closest to 0), reflecting cells pairs which did not encode reactivated content. (Similar results were obtained using the decile with the lowest signed contribution.)

Having established the reactivated and non-reactivated (baseline) cell pairs for each session, the reactivation content was identified by analysing when during the task the reactivated cell pairs were more coactive than the non-reactivated cell pairs. Coactivity was used for this measure for methodological consistency, because the (R)EV method depends on firing rate correlations between the cell pair: high EV-REV is driven by coherent fluctuations in firing rate (we ignore the possibility that synchronous decreases or pauses in firing rate might encode task-relevant information). To measure coactivity, the binless 50-ms spike trains for the two members of a cell pair were compared, and a pointwise minimum was taken between them such that if either cell had low or zero firing rate, the coactivity would be correspondingly low or zero. The coactivity was then z-scored with respect to the whole recording session to control for bias by the cells' inherent firing rates.

**Behavioural correlates of preferentially reactivated cell pairs.** With the hypothesis that reactivated CA1-striatal or striatal-striatal cell pairs preferentially encoded reward prediction and/or error, coactivity was compared between reactivated and non-reactivated cell pairs and between their coactivity on high- and medium-probability arms on the approach to the reward location (CA1-striatal) or after rewarded outcome (striatal-striatal). A nested mixed-effects ANOVA was constructed with cell-pair type (reactivated or non-reactivated) and arm (high or medium) as fixed effects, cell-pair identity nested within rat identity as random effects and mean z-scored coactivity of a cell pair in the 2 s prior to arrival at the reward location (for CA1-striatal pairs) or 5 s after arrival at the reward location (for striatal-striatal pairs, on rewarded trials only) as the dependent variable. The interaction between the two fixed effects was the effect of interest, with post-hoc $t$-tests conducted to compare coactivity between reactivated and non-reactivated cell pairs on each arm separately.

### Reporting summary
Further information on research design is available in the Nature Portfolio Reporting Summary linked to this article.

## Data availability
Pre-processed data used in this study is available at https://github.com/EmmaRoscow/QlearningReplay[112]. Source data for figures are provided with this paper. Source data are provided with this paper.

## Code availability
All code used in this study is available at https://github.com/EmmaRoscow/QlearningReplay[112].

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

## Acknowledgements

We are grateful to Aleksander Domanski and Andrew New for assistance with experimental set-up, and Luke Burguete for assistance with spike-sorting. E.L.R. received funding from a Wellcome Trust PhD scholarship (109070/Z/15/Z); M.W.J. received funding from a Wellcome Senior Research Fellowship in Basic Biomedical Science (202810/Z/16/Z).

## Author contributions

Conceptualisation, E.L.R., N.F.L. and M.W.J.; Methodology, E.L.R., N.F.L. and M.W.J.; Investigation, E.L.R.; Formal Analysis, E.L.R. and T.H.; Data Curation, E.L.R. and T.H.; Writing—Original Draft, E.L.R., N.F.L. and M.W.J.; Writing—Reviewing and Editing, E.L.R. and M.W.J.; Visualisation, E.L.R. and T.H.; Supervision, N.F.L. and M.W.J.; Funding Acquisition N.F.L. and M.W.J.

## Competing interests

The authors declare no competing interests.
