## [Transparent Peer Review file · Nature Communications]

Post-learning replay of hippocampal-striatal activity is biased by reward-prediction signals

Corresponding Author: Dr Emma Roscow

A version of this paper was originally rejected for publication by Nature Communications, however that decision was reconsidered after appeal by the authors.

Version 0:

Reviewer comments:

Reviewer #1

(Remarks to the Author)

In this manuscript, the authors test the contribution of sleep memory replay to a reinforcement learning model of animal behavior. The authors fit the model to the actual behavioral outcomes of rats that search for reward in a 3-arm maze in which each arm is associated with a different probability of obtaining reward. Next, they introduce replay of previous experience according to one of several “replay policies” and quantify the difference in fit between models with and without added replay. The authors conclude that a model with replay that is prioritized by reward prediction error (RPE) better fits observed behavior in rats than alternative models without replay, with random replay or with reward-biased replay.

The benefit of “experience replay” or “prioritized experience replay” (prioritized by temporal-difference error) was demonstrated previously in the context of neural network models of reinforcement learning (i.e. deep Q networks). The manuscript appears to corroborate these earlier results, although it finds a detrimental effect of replay policies that do not prioritize RPE – which is surprising. The results show that a single replay event per session is sufficient for the beneficial (or detrimental) effects in the model, which is also surprising.

The manuscript does not address if the RPE based replay policy matches what happens in the brain, which could be done by showing that hippocampal sleep replay in the rats that performed the task is RPE-prioritized. While the manuscript is conceptually interesting, the absence of an experimental demonstration limits the value of this work.

I have several major and minor concerns regarding the methods and presentation of the data that need to be addressed before the manuscript can be accepted for publication.

- The title says that there is “behavioural and computational evidence” for RPE-biased memory consolidation, however it is not clear to me how the behavioral results constitute evidence for this conclusion. I suggest removing “behavioural”.
- As mentioned above and as also recognized by the authors, the beneficial effects of (prioritized) experience replay in reinforcement learning models have been demonstrated before with neural network approaches (e.g. deep Q networks). The authors should explain in more detail the unique contributions of their work in comparison to this earlier work.
- The behavioral performance of the rats (figure 1) is compared to a chance level of 33%. However, rats often have a natural bias against a repeat visit to the same location and thus a more conservative chance level is 50%. What is the frequency of repeat visits in the dataset? The authors should test behavioral performance also against the more conservative 50% chance level. This is most important for the later sessions (e.g. reversal phase) at which time the rats have likely learned to make no (very few) repeat visits.
- How is the Q-matrix initialized and how do the initial Q values affect learning? Given the natural behavioral biases in rats (e.g. against repeat visits) at the start of the experiment, could a non-uniform Q-matrix initialization be used to speed up learning? Perhaps the initial Q-values should be parameters in the optimization (possibly with constraints to keep the number of parameters low)?

- In figure 2a, the value of the square error (0.33) for the Low action in both the table and the formula for the reliability error is incorrect.

- Figure 3a: it is not clear what each point in the scatter plot represents – a trial, an action or something else? The description in the legend suggests that the scatter plot should have colors, but it does not. The legend also talks about observed action frequencies being averaged over similar predicted action probabilities. Does this mean that each point is an average? How is “similar” defined in this case? Please clarify.

- Figure 3b: I think a scatter plot of the (absolute) error and session # (optionally with a distribution of the errors next to it, would more clearly convey the relation between the two variables.

- To understand the different replay policies, it would be very informative to illustrate which trials were selected for replay with an example. For example, the authors could show a timeline of all trials in an experiment (or of a few selected sessions) and indicate for each trial: the behavior (sequence of states and rewards), the reward-prediction error (RPE), replay frequency according to the various replay policies. It would also be helpful to include the evolution of Q-values in this figure (although perhaps not for every single trial).

- There are two surprising results in the data: first, that random and reward-biased replay policies have a negative impact on the model fit (not just no effect) and second, that few replay events can have such a large effect on the model fit. According to figure 4, most of the (positive and negative) effect is already obtained with a single replay event per session. From the data presented in the manuscript it is not possible to derive how the replay effect is manifested, e.g. whether it is due to a systematic change in one of the basic parameters (learning rate, discount and exploration) or a faster and more accurate construction of the Q-matrix. The better evaluate the replay effect, the authors should show optimal fitted parameters for all replay policies and quantitatively explore the differences in optimal parameters between replay policies. In addition, they should show the learned Q matrices and how they differ between replay policies (how much do they differ from the optimal policy for this task?). This is a crucial point.

- Line 228 the authors refer to figure 4d-e to support their claim that a higher number of replay events had a strong effect on improving reliability errors for RPE-prioritized and RPE-proportional replay policies. However, it is not clear how this conclusion is drawn from these figures and the effect should be properly quantified and statistically tested.

- Equations 1 and 2 on page 7 are identical to equations 3 and 4 in the methods and are redundant.

- The authors define a “reliability error” that is used as a cost function for parameter optimization and to compare the model forecasts to the observed behavior of the animals. I have several concerns about the definition and use of the reliability error.

The reliability error is based on a method to compare weather forecasts to actual weather patterns. It appears that the authors compute the error between the forecast probabilities at a single trial (computed from the Q matrix) and the average frequency of observed (state, action) pairs across all trials and sessions. It is not clear to me why minimizing the difference between the action probabilities at a single trial and the average behavior across all trials is the desired approach. Rather, one may want to minimize the difference between the estimated action probabilities at a single trial and the actual action taken (e.g. error = $([0.09, 0.47, 0.44] - [0, 1, 0])^2$, if the rat chose the middle arm). Could the authors explain why they have chosen the reliability error for parameter optimization and quantification of model fit, and why they think it is an appropriate measure?

Further, if I compare the work in the cited paper (Murphy and Murphy, 1973) and the method used by the authors, I do not see a 100% correspondence – although this may be my superficial understanding of the math. In the cited paper, the probability space is partitioned into distinct categories (e.g. [0, 0.1), [0.1, 0.2), etc.) and reliability is computed for each partition. This is different from what the authors appear to do. Although there is mention in the results of averaging “over similar predicted action probabilities” in the legend of figure 3 and binning of the predicted action probabilities to compute a correlation (page 8), the description in the methods do not make this clear. I am concerned that the reliability error that the authors defined is not correctly adapted from (Murphy and Murphy, 1973). Could the authors provide clarification?

On page 20, the reliability error per trial is defined (equation 5), but how is the overall reliability error (or cost) computed from the per-trial reliabilities to perform the parameter optimization?

Finally, the reliability error does not provide an intuitive measure to evaluate how well the models fit the behavior (what does a value of 10 really mean?). It would be helpful to simulate the behavior based on the fitted models and visualize this together with the actual behavior.

- Equation 7 (page 22): what does the small p in the denominator represent?

- The effect of replay on prediction accuracy is only measured at the population level. How consistency is the effect across subjects? Please provide data for individual animals.

- Is the phi parameter in equations 7 and 10 one and the same? If so, what is 5th parameter for RPE replay and if not, please use different symbols.

- Table 1 shows the optimal parameters for Q-learning models without replay for each subject. There appears to be a large spread across animals, especially for the alpha and gamma parameters. What does this mean? One concern is that the ability to correctly estimate parameter values in reinforcement learning models (i.e. model identifiability) is hard. The apparent correlation between the alpha and gamma parameters in table 1 may indicate the model is not or only partially identifiable.

- In the parameter optimization, the authors assume that parameter values are stationary across all sessions, which may not be the case. Ideally, changes of (some of) the parameters over time would be included in the model. If that is not feasible, the authors should at least discuss the impact of possible non-stationarity on their results.

- On page 23, the authors state that the data and code are available on github, however the repository is empty. I think it is great that the authors plan to make data and code publicly available eventually, but it would be really helpful for the reviewers to have access as part of the evaluation of the manuscript.

Reviewer #2

(Remarks to the Author)

In my opinion, this is an original and interesting study that highlights an important aspect of analysis of the "offline" neuronal activity. The authors rightly point out that typical experiments and/or data analyses often not distinguish between reward-driven behavior and other possible contexts and scenarios, e.g., a reward prediction error (RPE) driven behavior. In order to delineate between the reward- and RPE-driven scenarios, the authors propose a simple experimental paradigm that produces positive and negative RPE and use a reinforcement-learning estimator to assess the differences in interpreting the replay activity. They collect a significant volume of behavioral data from 6 rats performing a learning task, and quantify the correlations between the predicted and the actual behavior. The Q-learning model provides stable distinction between reward and PRE-based replay scenarios and leads the authors to conclude that the PRE-biased interpretation of the replay activity better explains the learning dynamics. The approach appears technically sound, data informed computational approach is conceptually valid and suitable for data analyses, and the results appear valid to me.

The mechanisms explaining the observed differences via dopaminergic neuromodulation (lines 49-52, 59-68 in the Introduction and then in the 3-4th paragraph in the Discussion) are physiologically plausible, but somewhat speculative. As far as I know, very few works, if any, address the issue of the replay scenarios directly. A theoretical discussion of differences in replay scenarios, but without referencing physiological mechanisms, is proposed in A. Babichev et al, "Replays of spatial memories suppress topological fluctuations in cognitive map," *Network Neuroscience* (2019)—this may help. Nevertheless, at the empirical level, I believe that the manuscript discusses an interesting finding.

Reviewer #3

(Remarks to the Author)

Roscow and colleagues present a computational model attempting at reproducing rat behavior in a multiple-choice, stochastic reward task, by way of reinforcement learning augmented with a "replay" phase. The replay phase was either random, with probability based on amount of obtained reward (as supported by some neurophysiological evidence) or based on reward-prediction error. RRP based replay gave the best improvement in the model power to predict animal behavior.

I think the idea is interesting, however the interest is limited by two factors

- Of course it would be nice to know whether replay is indeed modulated by RPE. The presence among the authors of a neurophysiologist quite active in the replay field makes it likely that they are working at that, and indeed that would change the value of the paper a lot. It is true that this stochastic reward task is especially suitable to test this idea.

- The reinforcement learning and reward prediction error is interesting, however I wonder whether the Q-learning model is a good fit for this task. Here, state-dependency is enforced by not rewarding repeated entry in an arm. This takes advantage of rodents' natural tendency to alternate between arms, and is typically thought to reflect working memory processes. In principle, the working memory component and the learning of the reward probability for each arm (akin to reference memory) could involve different brain processes and structures, so it may be a bit of a stretch to use the same learning process to model both.

In particular it would be interesting to have more detail on how different sorts of "errors" (deviation from the optimal policy of alternating between the high prob. and middle prob. arm) are reproduced by the model, for example working memory (repetition) errors vs. entries to the low prob. arm.

OTHER POINTS

- p. 11 "First, all replay policies showed improvements over no-replay in early sessions" Is this significant? From the figure the effect looks very small

- how is the forecast action probability calculated from the Q-values? By softmax? I couldn't find a formula. Doesn't that introduce another parameter to optimize?

Version 1:

Reviewer comments:

Reviewer #3

(Remarks to the Author)

The paper has improved in this revision, for example with the initialization that takes the natural tendency for alternation into account. I understand that this is a way to model this natural behavior without having to invoke other mechanism, and appreciate the simplicity of the approach.

The big new thing is of course the neural data, that are collected according to state of the art practice. The results are of interest, my main point would be to improve the way they are linked to the rest of the study. Namely, the model seems to make the key point that RPE-based replay is a better fit to the behavioral data than reward-based replay, however, the electrophysiological results shown (approach to the high-reward arm overrepresented in replay) would be compatible with both idea, or if anything, closer to the reward-based option. Would it be possible to look for something in the data that is closer to the RPE policy (perhaps looking at what happens after the reward is delivered or not)? I understand that there may be behavioral confounds (reward consumption etc.) but perhaps one could look for rewarded (or unrewarded) trials for high vs low expectation trials? The Q learning framework would probably have some predictions for those comparisons.

(Remarks on code availability)

It seems to be a complete collection of matlab scripts to reproduce the results presented in the paper

Version 2:

Reviewer comments:

Reviewer #3

(Remarks to the Author)

The authors made valuable additions to the paper in response to my comments. As it stands, the study is in my view suitable for publication.

(Remarks on code availability)

Reviewer #1 (Remarks to the Author):

In this manuscript, the authors test the contribution of sleep memory replay to a reinforcement learning model of animal behavior. The authors fit the model to the actual behavioral outcomes of rats that search for reward in a 3-arm maze in which each arm is associated with a different probability of obtaining reward. Next, they introduce replay of previous experience according to one of several “replay policies” and quantify the difference in fit between models with and without added replay. The authors conclude that a model with replay that is prioritized by reward prediction error (RPE) better fits observed behavior in rats than alternative models without replay, with random replay or with reward-biased replay.

The benefit of “experience replay” or “prioritized experience replay” (prioritized by temporal-difference error) was demonstrated previously in the context of neural network models of reinforcement learning (i.e. deep Q networks). The manuscript appears to corroborate these earlier results, although it finds a detrimental effect of replay policies that do not prioritize RPE – which is surprising.

Thanks to your helpful and insightful suggestion to initialise the Q-values to reflect rats' natural tendency to alternate between arms on a maze, we have completely re-run the fitting of the Q-learning parameters to the behavioural data and obtained a different set of results. The new results do not show a detrimental effect of policies that do not prioritise RPE (please see new Fig 5a), but RPE-prioritised replay remains the only form of replay generating a significantly better model of the rats' real learning.

The results show that a single replay event per session is sufficient for the beneficial (or detrimental) effects in the model, which is also surprising.

This has also changed in the new results based on your recommended approach to initialisation of Q-values. These new results show a more graduated, cumulative effect of replay events, although even one replay event does still have a statistically significant effect in the RPE-prioritised condition. One replay event represents a 2.2% increase in the number of Q-value updates compared to the average 45 trials per session, and we believe this is not out of proportion for the effect size for one replay event. We also note that experimental studies have shown that (1) sequence replays are relatively sparse (approximately 1 per 5min of non-REM sleep in rats, Lee & Wilson 2002, Neuron 36: 1183-1194) and (2) even brief stimulation of hippocampal synapses with place cell spiking patterns encompassing only a few replay events is sufficient to induce synaptic plasticity (Sadowski et al. 2016, Cell Reports <http://dx.doi.org/10.1016/j.celrep.2016.01.061>).

The manuscript does not address if the RPE based replay policy matches what happens in the brain, which could be done by showing that hippocampal sleep replay in the rats that performed the task is RPE-prioritized. While the manuscript is conceptually interesting, the absence of an experimental demonstration limits the value of this work.

We agree that our original submission would have been bolstered by neurophysiological data and, inspired by your suggestion and the comments of Reviewer 3, have now completed a set of simultaneous multiple single unit recordings from rat ventral striatum and dorsal CA1 (new Fig 10). In brief, we first show that a subset of both CA1 and vStr neurons ramp-up activity on approach to reward. We then show that pairs of vStr- vStrand CA1- vStr neurons that are coactive during task performance AND significantly reactivated during post-task rest (sharpwave-ripples) are preferentially recruited to encode this reward anticipation on the maze.

We thank you for your prompt to undertake this work and hope you agree that the results show that our model predictions may indeed translate to neurobiological mechanisms.

I have several major and minor concerns regarding the methods and presentation of the data that need to be addressed before the manuscript can be accepted for publication.

- The title says that there is “behavioural and computational evidence” for RPE-biased memory consolidation, however it is not clear to me how the behavioral results constitute evidence for this conclusion. I suggest removing “behavioural”.

In light of your comments and the additional neurophysiological data, we have updated the title to ‘Post-learning replay of hippocampal-striatal activity is biased by reward-prediction errors’.

- As mentioned above and as also recognized by the authors, the beneficial effects of (prioritized) experience replay in reinforcement learning models have been demonstrated before with neural network approaches (e.g. deep Q networks). The authors should explain in more detail the unique contributions of their work in comparison to this earlier work.

Our most important unique contributions are (1) showing the link to actual animal behaviour in this integrated study, and (2) the new electrophysiological data now specifically implicating CA1- vStr circuitry in the effects predicted by the models.

- The behavioral performance of the rats (figure 1) is compared to a chance level of 33%. However, rats often have a natural bias against a repeat visit to the same location and thus a more conservative chance level is 50%. What is the frequency of repeat visits in the dataset? The authors should test behavioral performance also against the more conservative 50% chance level. This is most important for the later sessions (e.g. reversal phase) at which time the rats have likely learned to make no (very few) repeat visits.

This is a very astute point, thank you – we have now set the chance level to 50% for our statistical analyses (see Fig 1c-d).

- How is the Q-matrix initialized and how do the initial Q values affect learning? Given the natural behavioral biases in rats (e.g. against repeat visits) at the start of the experiment, could a non-uniform Q-matrix initialization be used to speed up learning? Perhaps the initial Q-values should be parameters in the optimization (possibly with constraints to keep the number of parameters low)?

Thank you again, as noted above we have now initialised the Q-values to do this and re-run the model.

- In figure 2a, the value of the square error (0.33) for the Low action in both the table and the formula for the reliability error is incorrect.

Apologies, this was a rounding error – but the re-calculated values are different now anyway.

- Figure 3a: it is not clear what each point in the scatter plot represents – a trial, an action or something else? The description in the legend suggests that the scatter plot should have colors, but it does not. The legend also talks about observed action frequencies being averaged over similar predicted action probabilities. Does this mean that each point is an average? How is “similar” defined in this case? Please clarify.

Apologies for the mistake. The legend has been amended accordingly, and explains that points are averages over a percentile.

- Figure 3b: I think a scatter plot of the (absolute) error and session # (optionally with a distribution of the errors next to it, would more clearly convey the relation between the two variables.

We did trial your suggested approach (see below), but retain our preference for the original histogram representation.

- To understand the different replay policies, it would be very informative to illustrate which trials were selected for replay with an example. For example, the authors could show a timeline of all trials in an experiment (or of a few selected sessions) and indicate for each trial: the behavior (sequence of states and rewards), the reward-prediction error (RPE), replay frequency according to the various replay policies. It would also be helpful to include the evolution of Q-values in this figure (although perhaps not for every single trial).

Thank you for these useful suggestions. New figures 4 and 7 show selection of trials for replay related to the four different replay policies, and new figure 6 exemplifies the evolution of Q-values across learning for an individual rat.

- There are two surprising results in the data: first, that random and reward-biased replay policies have a negative impact on the model fit (not just no effect) and second, that few replay events can have such a large effect on the model fit. According to figure 4, most of the (positive and negative) effect is already obtained with a single replay event per session. From the data presented in the manuscript it is not possible to derive how the replay effect is manifested, e.g. whether it is due to a systematic change in one of the basic parameters (learning rate, discount and exploration) or a faster and more accurate construction of the Q-matrix. To better evaluate the replay effect, the authors should show optimal fitted parameters for all replay policies and quantitatively explore the differences in optimal parameters between replay policies. In addition, they should show the learned Q matrices and how they differ between replay policies (how much do they differ from the optimal policy for this task?). This is a crucial point.

Neither of these effects appear in the new results, but figure 6 now shows the learned Q-values and how they differ by replay policy.

- Line 228 the authors refer to figure 4d-e to support their claim that a higher number of replay events had a strong effect on improving reliability errors for RPE-prioritized and RPE-proportional replay policies. However, it is not clear how this conclusion is drawn from these figures and the effect should be properly quantified and statistically tested.

Apologies for this omission, these statistics now reported in figure 5a and the associated Results text.

- Equations 1 and 2 on page 7 are identical to equations 3 and 4 in the methods and are redundant.

Apologies, these equations have now been amended.

- The authors define a “reliability error” that is used as a cost function for parameter optimization and to compare the model forecasts to the observed behavior of the animals. I have several concerns about the definition and use of the reliability error.

The reliability error is based on a method to compare weather forecasts to actual weather patterns. It appears that the authors compute the error between the forecast probabilities at a single trial (computed from the Q matrix) and the average frequency of observed (state, action) pairs across all trials and sessions. It is not clear to me why

minimizing the difference between the action probabilities at a single trial and the average behavior across all trials is the desired approach. Rather, one may want to minimize the difference between the estimated action probabilities at a single trial and the actual action taken (e.g. error = $([0.09, 0.47, 0.44] - [0, 1, 0])^2$, if the rat chose the middle arm). Could the authors explain why they have chosen the reliability error for parameter optimization and quantification of model fit, and why they think it is an appropriate measure?

The reason is that the animals' behaviour is probabilistic, so any model which predicts a deterministic $[0, 1, 0]$ is necessarily poorly calibrated. A cost function based on this would reward over-confidence in the model, for example if arm 1 is chosen more often than the other arms, a "solution" which tends to always predict arm 1 with high probability will get a good score; but this is a bad model of the probabilistic behaviour of the rat. This is especially difficult because there is a disparity in how often each arm is chosen. Ideally if the rat chose one arm but another arm was a close second, we want a model which reproduces not just the winning arm but the close second as well. We don't have access to that from just one trial outcome (we only know what the winning arm was), so we need to combine all of the trials.

Further, if I compare the work in the cited paper (Murphy and Murphy, 1973) and the method used by the authors, I do not see a 100% correspondence – although this may be my superficial understanding of the math. In the cited paper, the probability space is partitioned into distinct categories (e.g. $[0, 0.1)$, $[0.1, 0.2)$, etc.) and reliability is computed for each partition. This is different from what the authors appear to do. Although there is mention in the results of averaging "over similar predicted action probabilities" in the legend of figure 3 and binning of the predicted action probabilities to compute a correlation (page 8), the description in the methods do not make this clear. I am concerned that the reliability error that the authors defined is not correctly adapted from (Murphy and Murphy, 1973). Could the authors provide clarification?

You are correct that the "cost function" we employed is adapted from what Murphy & Murphy call reliability. We have renamed this the "error score" throughout the manuscript to avoid implication that we have used Murphy & Murphy's approach verbatim.

On page 20, the reliability error per trial is defined (equation 5), but how is the overall reliability error (or cost) computed from the per-trial reliabilities to perform the parameter optimization?

We have now clarified on p27 ('Parameter-fitting for Q-learning' section) that this is averaged over runs.

Finally, the reliability error does not provide an intuitive measure to evaluate how well the models fit the behavior (what does a value of 10 really mean?). It would be helpful to simulate the behavior based on the fitted models and visualize this together with the actual behavior.

Figure 6 of the revised manuscript gives a visual comparison of behaviour alongside modelled Q-values, which are proportional to action predictions, which we hope reinforces intuition.

- Equation 7 (page 22): what does the small p in the denominator represent?

This was a typographical error and has been amended (p28).

- The effect of replay on prediction accuracy is only measured at the population level. How consistency is the effect across subjects? Please provide data for individual animals.

This is a good point, and the effect per subject is now reported on p11 (lines 241-243).

- Is the ϕ parameter in equations 7 and 10 one and the same? If so, what is 5th parameter for RPE replay and if not, please use different symbols.

Thank you for spotting this, these now have different symbols.

- Table 1 shows the optimal parameters for Q-learning models without replay for each subject. There appears to be a large spread across animals, especially for the alpha and gamma parameters. What does this mean? One concern is that the ability to correctly estimate parameter values in reinforcement learning models (i.e. model identifiability) is hard. The apparent correlation between the alpha and gamma parameters in table 1 may indicate the model is not or only partially identifiable.

The spread of these values is smaller now we have adopted your suggested approach to optimising parameter values.

- In the parameter optimization, the authors assume that parameter values are stationary across all sessions, which may not be the case. Ideally, changes of (some of) the parameters over time would be included in the model. If that is not feasible, the authors should at least discuss the impact of possible non-stationarity on their results.

We have added a comment acknowledging this simplifying assumption on p10 (lines 206-208): *“The model makes a simplifying assumption of stationary parameters throughout learning, which may deviate from biological reality (Coddington et al. 2023) but prioritises interpretability of the fitted parameter values and prevents overfitting to an overly complex model.”*

- On page 23, the authors state that the data and code are available on github, however the repository is empty. I think it is great that the authors plan to make data and code publicly available eventually, but it would be really helpful for the reviewers to have access as part of the evaluation of the manuscript.

The data and code are now available at the specified Github repo.

Reviewer #2 (Remarks to the Author):

In my opinion, this is an original and interesting study that highlights an important aspect of analysis of the “offline” neuronal activity. The authors rightly point out that typical experiments and/or data analyses often not distinguish between reward-driven behavior and other possible contexts and scenarios, e.g., a reward prediction error (RPE) driven behavior. In order to delineate between the reward- and RPE-driven scenarios, the authors propose a simple experimental paradigm that produces positive and negative RPE and use a reinforcement-learning estimator to assess the differences in interpreting the replay activity. They collect a significant volume of behavioral data from 6 rats performing a learning task, and quantify the correlations between the predicted and the actual behavior. The Q-learning model provides stable distinction between reward and PRE-based replay scenarios and leads the authors to conclude that the PRE-biased interpretation of the replay activity better explains the learning dynamics. The approach appears technically sound, data informed computational approach is conceptually valid and suitable for data analyses, and the results appear valid to me.

*The mechanisms explaining the observed differences via dopaminergic neuromodulation (lines 49-52, 59-68 in the Introduction and then in the 3-4th paragraph in the Discussion) are physiologically plausible, but somewhat speculative. As far as I know, very few works, if any, address the issue of the replay scenarios directly. A theoretical discussion of differences in replay scenarios, but without referencing physiological mechanisms, is proposed in A. Babichev et al, “Replays of spatial memories suppress topological fluctuations in cognitive map,” *Network Neuroscience* (2019)—this may help. Nevertheless, at the empirical level, I believe that the manuscript discusses an interesting finding.*

Thank you for these supportive comments and for highlighting the Babichev paper, which we have now cited in the Discussion. As noted in our responses to Reviewers 1 and 3, we have now included neurophysiological evidence that RPE-biased replay is instantiated in hippocampal-ventral striatal circuits (Fig 10), extending our suggested links to dopaminergic signalling beyond speculation.

Reviewer #3 (Remarks to the Author):

Roscow and colleagues present a computational model attempting at reproducing rat behavior in a multiple-choice, stochastic reward task, by way of reinforcement learning augmented with a "replay" phase. The replay phase was either random, with probability based on amount of obtained reward (as supported by some neurophysiological evidence) or based on reward-prediction error. RRP based replay gave the best improvement in the model power to predict animal behavior.

I think the idea is interesting, however the interest is limited by two factors

- Of course it would be nice to know whether replay is indeed modulated by RPE. The presence among the authors of a neurophysiologist quite active in the replay field makes it likely that they are working at that, and indeed that would change the value of the paper a lot. It is true that this stochastic reward task is especially suitable to test this idea.

Reviewer 1 also voted for neurophysiological testing of our model predictions. As noted in our responses above, we have now undertaken a set of simultaneous multiple single unit recordings from rat ventral striatum and dorsal CA1 (new Fig 10). In brief, we first show that a subset of both CA1 and VS neurons ramp-up activity on approach to reward on the 3-armed maze. We then show that pairs of VS-VS and CA1-VS neurons that are coactive during task performance AND significantly reactivated during post-task rest (during periods encompassing sharpwave-ripples) are preferentially recruited to encode this reward anticipation on the maze. We hope you enjoy these data!

- The reinforcement learning and reward prediction error is interesting, however I wonder whether the Q-learning model is a good fit for this task. Here, state-dependency is enforced by not rewarding repeated entry in an arm. This takes advantage of rodents' natural tendency to alternate between arms, and is typically thought to reflect working memory processes. In principle, the working memory component and the learning of the reward probability for each arm (akin to reference memory) could involve different brain processes and structures, so it may be a bit of a stretch to use the same learning process to model both.

This is an interesting point, and we completely agree that the field at large should work towards RL-based models integrating a range of psychological processes, including working memory. Rather than derive such a model, we have chosen to focus on the neurophysiological testing of our more tractable, Q-learning approach. Your concerns are also somewhat mitigated by our updated method, which initialises the Q-values to reflect the rats' tendency to alternate between arm choices.

In particular it would be interesting to have more detail on how different sorts of "errors" (deviation from the optimal policy of alternating between the high prob. and middle prob. arm) are reproduced by the model, for example working memory (repetition) errors vs. entries to the low prob. arm.

This detail has been added on p8 (lines 184-186).

OTHER POINTS

- p. 11 "First, all replay policies showed improvements over no-replay in early sessions" Is this significant? From the figure the effect looks very small

Apologies for this omission; significance levels have now been marked on Fig 5a and are noted in associated text.

- how is the forecast action probability calculated from the Q-values? By softmax? I couldn't find a formula. Doesn't that introduce another parameter to optimize?

Apologies once more, this detail has been added to materials and methods (equation 3) and depends on the epsilon parameter, as discussed in our response to Review 1's first point.

Roscow et al., Response to reviewer

Reviewer #3 (Remarks to the Author):

The paper has improved in this revision, for example with the initialization that takes the natural tendency for alternation into account. I understand that this is a way to model this natural behavior without having to invoke other mechanism, and appreciate the simplicity of the approach.

The big new thing is of course the neural data, that are collected according to state of the art practice. The results are of interest, my main point would be to improve the way they are linked to the rest of the study. Namely, the model seems to make the key point that RPE-based replay is a better fit to the behavioral data than reward-based replay, however, the electrophysiological results shown (approach to the high-reward arm overrepresented in replay) would be compatible with both idea, or if anything, closer to the reward-based option.

Would it be possible to look for something in the data that is closer to the RPE policy (perhaps looking at what happens after the reward is delivered or not)? I understand that there may be behavioral confounds (reward consumption etc.) but perhaps one could look for rewarded (or unrewarded) trials for high vs low expectation trials? The Q learning framework would probably have some predictions for those comparisons.

Reviewer #3 (Remarks on code availability):

It seems to be a complete collection of matlab scripts to reproduce the results presented in the paper

Authors' response:

We are very grateful for your willingness to revisit this manuscript and for your further helpful comments – thank you. These comments have inspired new analyses, unveiling additional neurophysiological evidence for RPE-biased reactivation and suggesting complementary roles for hippocampal-striatal vs. intra-striatal reactivation.

You are correct in noting that our initial analyses of hippocampal-ventral striatal reactivation fell short of direct alignment with our reinforcement learning model predictions. We previously showed that reactivated hippocampal-ventral striatal cell pairs ramped up coactivity on approach to reward (as opposed to signalling receipt or absence of reward – this has been reinforced by an analysis of rewarded vs. unrewarded trials in Supplementary Fig S1); that expectation signal was related to, but not equivalent to, an RPE signal. Our interpretation at the time was that our ventral striatal recordings were one step removed (anatomically and computationally) from VTA-based RPE signals.

However, your latest comments have inspired an important addition: alongside analysing the behavioural correlates of reactivated CA1-ventral striatal cell pairs, we have now analysed striatal-striatal pairs (which are also significantly reactivated, as shown in Fig 10d). During the task, these intra-striatal pairs also ramp up coactivity on approach to reward, but significantly more so on approach to less certain reward (medium probability arm trials) than more certain reward (high probability arm trials, new Fig 10j-k). This preferential coactivation on approach to less certain reward – coupled with subsequent enhanced coactivity during reward outcome (Fig 10j, left panel) – brings the data closer to the RPE policy predicted by the Q-learning model.

Again, thank you for your time and thoughtful suggestions.

Updates to the manuscript include:

- Minor changes to the Abstract and Introduction to accommodate new results, lines 19, 65-66, and 95-101
- Minor clarifying changes to the Results, lines 107-109, and 327-328, referencing an additional paper
- Added a new analysis of CA1-vStr reactivation, showing that cell pairs including a significantly reward-modulated striatal unit are more likely to be reactivated, lines 357-361
- Added a supplementary figure showing that including rewarded trials only gives qualitatively similar results to all trials, consistent with reward per se not being the key driver of hippocampal-striatal coactivity (separate supplementary file, and referred to on lines 372-373)
- Striatal reactivation: two new panels added to the electrophysiology figure, 10j-k, showing results of intra-striatal reactivation. The results are described on lines 376-387.
- Slight modifications to the discussion:
 - Lines 395-396
 - Line 422
 - Lines 425-429
 - Lines 432-438
 - Line 535
 - Line 538
- More substantial modifications to the discussion:
 - Lines 441-453, discussing why there appears to be different replay content in our recordings, and referencing 2 additional papers
- Very minor changes to the materials and methods on lines 754, 778-779, and 784-785 regarding new analyses
- Last but not least, an updated title, reflecting the nuanced differences between hippocampal-striatal and intra-striatal reward signalling: **'Post-learning replay of hippocampal-striatal activity is biased by reward-prediction signals'**